# Hyperparameter Sensitivity in Deep Outlier Detection
## Analysis and a Scalable Hyper-Ensemble Solution

**Xueying Ding**
Carnegie Mellon University
`xding2@cs.cmu.edu`

**Lingxiao Zhao**
Carnegie Mellon University
`lingxiao@cmu.edu`

**Leman Akoglu**
Carnegie Mellon University
`lakoglu@cs.cmu.edu`

## Abstract

Outlier detection (OD) literature exhibits numerous algorithms as it applies to diverse domains. However, given a new detection task, it is unclear how to choose an algorithm to use, nor how to set its hyperparameter(s) (HPs) in unsupervised settings. HP tuning is an ever-growing problem with the arrival of many new detectors based on deep learning, which usually come with a long list of HPs. Surprisingly, the issue of model selection in the outlier mining literature has been "the elephant in the room"; a significant factor in unlocking the utmost potential of deep methods, yet little said or done to systematically tackle the issue. In the first part of this paper, we conduct the first large-scale analysis on the HP sensitivity of deep OD methods, and through more than 35,000 trained models, quantitatively demonstrate that model selection is inevitable. Next, we design a HP-robust and scalable deep *hyper-ensemble* model called ROBOD that assembles models with varying HP configurations, bypassing the choice paralysis. Importantly, we introduce novel strategies to speed up ensemble training, such as parameter sharing, batch/simultaneous training, and data subsampling, that allow us to train fewer models with fewer parameters. Extensive experiments on both image and tabular datasets show that ROBOD achieves and retains robust, state-of-the-art detection performance as compared to its modern counterparts, while taking only 2-10% of the time by the naïve hyper-ensemble with independent training.

## 1  Introduction

Outlier detection (OD) finds numerous real-world applications in finance, security, healthcare, to name a few. Thanks to this popularity, the literature has grown to offer a large catalog of detection algorithms [1]. With the recent advances in deep learning, the literature has been booming with the addition of many more OD models based on deep neural networks (NNs). (See surveys [9, 31, 36].)

While there is no shortage of OD methods today, given a new task, it is unclear how to choose which algorithm or model to use, nor how to configure its hyperparameter(s) (HPs) in unsupervised settings. That is, the fundamental problem of outlier model selection remains vastly understudied. Several evaluation studies have illustrated the sensitivity to HPs for traditional (i.e. non-deep) OD methods [2, 8, 17]. Most surprisingly, however, the issue of HP tuning/model selection for the newly burgeoning deep OD models has been "the elephant in the room"; a well-known problem that no one seems to want to bring up, which is exactly the focus of this paper.

Deep OD models are promising thanks to appealing properties such as task-driven representation learning and end-to-end optimization. On the other hand, while their traditional counterparts had only 1-2 HPs[1], deep OD models come with a long list of HPs: ($i$) architecture HPs (e.g. depth, width), ($ii$) regularization HPs (e.g. dropout, weight decay rates), and ($iii$) optimization HPs (e.g.

---

[1]e.g., $k$ in nearest neighbor based LOF [7], $\nu$ in OCSVM [39], sample size $\psi$ and #trees $t$ in IF [28].

36th Conference on Neural Information Processing Systems (NeurIPS 2022).

learning rate, epochs). These are inherited ones from regular deep NNs while most also exhibit their model-specific/specialized HPs. It would not be a freak occurrence to assume that their performance is heavily dependent on these HP settings. However, our closer analysis of the experiment testbeds in recent literature on deep OD models falls far from systematically addressing the issue. (See Appx. A.1 Table 8 for a preview.) We find hardly any discussion on model selection, with only a few work empirically studying sensitivity but to model-specific HPs only. Majority of work report results for a single "recommended" (how, unclear) configuration used for all datasets, or tune only a subset of the HPs on labeled validation and sometimes even test (!) data. To the best of our knowledge, there is no existing work that attempts (unsupervised) model selection for deep OD models.

In this work, our research goals are two-fold. **First**, through extensive experiments, we quantitatively demonstrate that deep OD models from various families are all sensitive to their HP settings. Our analysis shows that model selection is inevitable and is key to truly unlock the utmost potential of deep OD models. **Second**, motivated by our analysis, we propose a scalable deep *hyper-ensemble* called ROBOD that obviates HP selection through assembly of deep autoencoders with varying HP configurations. To speed up ensemble training, we introduce novel architectural and training strategies, and train fewer models, with fewer parameters, on smaller subsamples of data; by leveraging parameter sharing and joint/simultaneous training. The main contributions of our work are as follows.

- **First large-scale study on HP sensitivity of deep OD models:** We build a large testbed to systematically measure the performance variability of deep OD models under varying HP settings. Our study involves models from four different families, on both image and vector data, under both "clean" (i.e. inlier only) as well as "polluted" (i.e. outlier-contaminated) training data, over 3 random initializations, 80-800+ different HP configurations per model across 4-8 unique HPs. Overall, our analysis involves more than 35,000 runs. (Sec. 3)
- ROBOD**, a new deep hyper-ensemble OD model:** Motivated by our empirical study, we propose a hyper-ensemble model called ROBOD which combines scores from a collection of models, each trained with a different HP configuration. Rather than trying to choose, ROBOD fully bypasses the choice of and hence sensitivity to HP settings, and achieves robust (i.e. stable) performance across different initializations. (Sec. 4.1)
- **Design strategies to speed up hyper-ensemble training:** We propose speed-up strategies to efficiently hyper-ensemble model depth and width. We use an autoencoder (AE) with skip-connections to simultaneously train multiple AEs with different depths. In addition, we employ batch training of multiple models and use zero-masking on shared parameters to get different widths. Together, these provide a 90-98% savings in running time. (Sec. 4.2)
- **Extensive experiments:** Besides our large-scale measurement study, we also perform experiments on additional benchmark datasets, comparing ROBOD to baseline deep OD models as well as a traditional tree-ensemble. ROBOD achieves competitive or often better performance which, importantly, exhibits low variance by random initialization. (Sec. 5)

We expect that our work will increase awareness and help shift the community's focus (at least to some extent) from building the next yet-another deep OD model toward the fundamental issue of unsupervised model selection and hyperparameter-robust model design. To foster future research, we open source all code and datasets at `https://github.com/xyvivian/ROBOD`.

## 2   Related Work

**Unsupervised Outlier Detection (OD).** There exists a large pool of what-is-now-called traditional, i.e. not deep learning based, OD methods [1, 10]. These methods work with the original features or subspaces thereof, and typically exhibit just one or two hyperparameters (HPs).[1] With recent advances in deep learning, there has been a boom in deep OD models, as those can learn new task-dependent feature representations and directly optimize an OD objective. Despite their short history, multiple surveys have been published that aim to cover this fast-growing literature [9, 31, 36]. While deep OD models have been shown to outperform their traditional counterparts, they exhibit a much longer list of (typically 4-8) HPs (e.g. depth, width, dropout, weight decay, learning rate, epochs, etc. besides other model-specific HPs) that makes them very challenging to tune in unsupervised settings.

**Model Selection in Unsupervised OD: Prior (Black) Art.** At large, unsupervised outlier model selection remains to be a vastly understudied, yet extremely important area. Various evaluation studies have reported traditional detectors to be quite sensitive to their HP choices [2, 8, 17], raising concern for the fair evaluation and comparison of different models. Earlier work on automatically selecting

HPs are limited to one-class models [14, 40, 41]. More recently, general-purpose *internal* (i.e., unsupervised) model evaluation heuristics have been proposed [15, 29, 30], which solely rely on the input data (without labels) and the output (i.e., outlier scores). MetaOD [47] employs meta-learning to transfer information from similar historical tasks to a new task for model selection, which has only been tested on traditional OD models. Different from those that aim to select a single model, ensemble models have also been employed for OD [3], including those that combine models from the same family [26] as well as heterogeneous detectors from different families [33].

Regarding deep OD methods, we have surveyed a large collection of recent papers and their experimental testbed and HP settings, a summary of which is given in Appx. A.1 Table 8. To our surprise, we found *hardly any discussion on model selection*, with only a few work presenting sensitivity analysis with respect to not all but some, model-specific HPs. While some work reserve labeled validation/hold-out data to tune a subset of the HPs [6, 18, 23], majority of them fix the HP values and call them "recommended"/default settings [5, 11, 37, 38, 46, 49]. Moreover, a non-negligible number of existing work choose some critical HPs empirically on test data (!) to yield optimum results [4, 35, 48] (See Table 8, last column). Some work that builds on previous models (e.g., deep SVDD-based methods [35] vs. multi-sphere extension [13], transformation-based methods [16] for images vs. their extension to vector data [5], AnoGAN [37] and the follow-up EGBAD [46]) use the same architecture and HP settings as the prior work for consistent/"fair" comparison. However it is unlikely that the same HP values would work comparably for different models.

Admittedly, it is challenging to tune (a long list of) HPs in the absence of labels, yet, the opacity in the deep OD literature warrants careful investigation on the stability of model performance under varying HP settings, and ultimately on the fair comparison between these and traditional OD methods.

**Deep Model Ensembles.** Recently, deep NN predictions have been found to be often poorly calibrated [20]. As Bayesian learning does not offer straightforward training, deep ensemble models have been proposed as a simple alternative [25] to improve predictive uncertainty, as well as efficient ways of training deep NN ensembles [19, 42]. In this work, we leverage ensemble modeling toward a different goal: to improve the stability and robustness of unsupervised OD models to HP settings, combining predictions from models with different HPs into an OD *hyper*-ensemble. The closest to our work is Wenzel *et al.*'s deep hyper-ensemble [43], which, different from ours, considers *supervised* problems, to further foster diversity in the ensemble and thereby achieve better uncertainty estimation.

## 3 Hyperparameter-Sensitivity Analysis of Deep OD

### 3.1 Testbed Setup

**Models.** We study HP sensitivity of five deep OD methods of four different types: a basic deep autoencoder VanillaAE trained with reconstruction loss, *robust* deep autoencoder RDA [48], *one-class* classification based DeepSVDD [35], *adversarial* training based GANomaly [4], and an (AE) *ensemble* model RandNet [11]. These exhibit 4 to 8 HPs, as listed in Table 1. (See Appx. A.2 for descriptions.) (Note that RandNet is **not** a **hyper**-ensemble: members use the same HP configs except for NN sparsity.) We define a grid of 2-3 different values for each HP, including the author-recommended values when available (See details in Appx. Table 9), and train each deep OD method with all combinations; yielding 81-864 different models. (Note that 192 RandNet models each consists of an ensemble of 50 or 200 AEs.) We repeat each experiment 3 times with different initializations.

Table 1: Deep OD models used for studying hyperparameter sensitivity. We give the number (in parenthesis) and the list of HPs for each method, along with the total number of models trained for evaluation. (See Appx. A.2 for HP descriptions and Appx. Table 9 for list of grid values per HP.)

| Method | List of hyperparameters (HPs) | #models |
|---|---|---|
| VanillaAE | (4) n_layers · layer_decay · LR · iter | 81 |
| RDA [48] | (6) $\lambda$ · n_layers · layer_decay · LR · inner_iter · iter | 324 |
| DeepSVDD [35] | (8) conv_dim · fc_dim · Relu_slope · pretr_iter · pretr_LR · iter · LR · wght_dc | 864 |
| GANomaly [4] | (6) $w_{adv} = 1$ · $w_{con}$ · $w_{enc}$ · z_dim · LR · iter | 162 |
| RandNet [11] | (8) n_layers · layer_decay · sample_r · ens_size · pretr_iter$= 100$ · iter · LR · wght_dc$= 0$ | 192 |

**Train/Test settings.** In their original papers, DeepSVDD and GANomaly are trained on what we refer to as Clean (inlier only) data, and tested on a disjoint test dataset. In contrast, RDA and RandNet consider the *transductive* setting where the train data is the same as the test data, containing inliers as

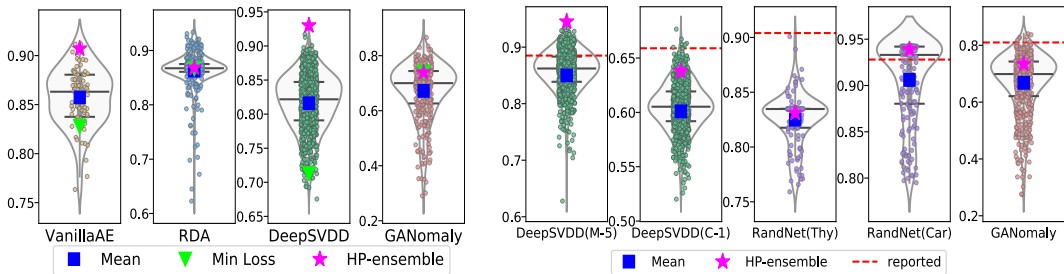

Figure 1: (left) AUROC performance of deep OD methods with different HP configurations (circles) on MNIST-4 showcase notable variation (i.e., sensitivity). (See footnote 2 below.) (right) Similar results on additional datasets; DeepSVDD on MNIST-5 and CIFAR10-auto, RandNet on Thyroid and Cardio, and GANomaly on MNIST-4out. Reported values (dashed lines) are overestimates of Mean (□). Hyper-ensemble (☆) improves notably over Mean.

well as outliers, which we refer to as Polluted. It is often understood to be more challenging than the Clean setting for unsupervised OD. For completeness, we evaluate these methods under both settings.

**Datasets.** For evaluation we consider both image point datasets. As in the original papers [4, 35], we use MNIST and CIFAR10 to construct OD tasks, except for RandNet which uses fully-connected rather than convolutional layers and is originally tested only on point-cloud benchmarks.

**Metrics.** Across all HP settings of a deep OD method, we report the mean AUROC, standard deviation (stdev), as well as minimum and maximum. Mean corresponds to the *expected* performance when a HP config is randomly picked from our (multi-dimensional) grid. We also report the average mean and its stdev over 3 repeated runs. For datasets that are directly comparable to those in the original papers, we contrast the mean model performance that we obtain to that reported value for author-recommended HPs. We compare to a simple model (i.e. HP) selection heuristic, which selects the (one) model with the lowest loss/objective value. In addition, instead of selecting one, we average the scores from *all* configurations and report the AUROC of this hyper-ensemble.

Over five deep OD methods, tens to hundreds of HP configs, multiple initializations, Clean and Polluted settings, and various datasets, we have trained a total of more than 35,000 models. As such, our study constitutes the first large-scale HP sensitivity analysis in the deep outlier mining literature.

## 3.2 Results and Observations

Fig. 1(left) provides the AUROC performances[2] for VanillaAE, RDA, DeepSVDD and GANomaly across all HP configurations (circles), for the Polluted setting (See Appx. A.3 Fig. 6 for Clean setting.) on MNIST-4 dataset where digit '4' images are designated as inliers and the rest nine classes are down-sampled at 10% as outliers, as in the DeepSVDD paper [35]. Horizontal bars mark the 1st, 2nd (i.e. median), and 3rd quartiles, square the mean AUROC across HPs, triangle the selected model by lowest loss, and star the AUROC of the hyper-ensemble. (Corresponding plots over 3 runs per method are in Appx. A.3 Fig. 7.) Table 2 provides summary statistics as well as the hyper-ensemble performance for comparison for both the (left) Clean and (right) Polluted settings.

First, in Fig. 1 we observe that all methods exhibit notable variability in performance across HPs, with many worse-than-average configurations; e.g. for GANomaly, as well as many better-than-average models; e.g. for RDA. Mean performance is considerably lower than the best model's, illustrating the opportunity or room for improvement. As one would expect, the performances in Table 2 are lower in the Polluted setting as compared to Clean (less so for the robust RDA), while sensitivity (i.e. stdev) is comparable or only slightly higher; showing that HP choice is critical under both settings.

We also find that selecting a model by the value of loss (▽), despite requiring to train and choose from all models, often is much worse than the mean, i.e. random picking a (single) model (□); proving this simple heuristic ineffective. On the other hand, the hyper-ensemble (☆) outperforms the mean in all cases, with quite small standard deviation across runs (i.e. low sensitivity to initialization).

---

[2]We remark that although the methods are presented side-by-side, the goal here is **not** to compare them "head-to-head" but rather, to analyze each one's performance variability by varying HPs on individual datasets.

Table 2: Basic stats of AUROC (%) distribution over varying HPs on MNIST-4 under (left) Clean and (right) Polluted settings. There is significant gap between the best and worst HP settings, with notable stdev around the mean. Polluted results have lower mean, and comparable or slightly higher variance. Hyper-ensemble outperforms random choice (i.e. mean), with low variability by initialization.

| Method | Min&Max | | Mean&Std. | Mean (avg. 3 runs) | Hyper-ens. Mean (avg. 3 runs) | Min&Max | | Mean&Std. | Mean (avg. 3 runs) | Hyper-ens. Mean (avg. 3 runs) |
|---|---|---|---|---|---|---|---|---|---|---|
| Vanilla AE | 87.41 | 98.19 | 93.12±2.91 | **93.12**±0.008 | **95.30**±0.03 | 76.34 | 91.20 | 85.73±2.95 | **85.76**±0.08 | **90.46**±0.12 |
| RDA | 81.79 | 95.96 | 88.94±2.43 | **88.95**±0.02 | **89.39**±0.02 | 62.29 | 93.35 | 86.30±3.68 | **86.24**±0.06 | **86.82**±0.04 |
| DeepSVDD | 75.75 | 96.58 | 92.39±2.02 | **92.37**±0.02 | **95.94**±0.05 | 67.54 | 91.71 | 81.65±4.15 | **81.66**±0.10 | **93.07**±0.70 |
| GANomaly | 17.49 | 95.73 | 78.90±16.63 | **78.26**±0.51 | **86.87**±0.31 | 29.06 | 86.60 | 67.20±10.30 | **67.45**±0.32 | **73.87**±0.52 |

The reported performance ($0.949 \pm 0.008$) of DeepSVDD on MNIST-4 (under Clean) [35] is similar yet somewhat optimistic over the mean value we obtain. MNIST-4 is an easy task for DeepSVDD since mean AUROC is already around $0.924$. Thus, we set up MNIST-5 (with digit '5' as inliers) with reported AUROC $0.885 \pm 0.009$, as well as CIFAR10-auto (with class 'automobile' as inliers) (in both cases rest of the classes are subsampled at 10% each as outliers) with reported AUROC $0.659 \pm 0.021$. We run DeepSVDD on both datasets under all 864 configurations. As shown in Fig. 1 (right), our mean AUROCs are $0.857 \pm 0.037$ for MNIST-5, and $0.605 \pm 0.024$ for CIFAR10-auto. (See Appx. A.3 Fig. 8 for all 3 runs.) The reported performances for the "recommended" HPs[3] in DeepSVDD (dashed red lines) appear to be optimistic over the mean, i.e. what one would expect by random choice (in the absence of any labels or other strategies).

The optimistic reporting trend holds for GANomaly and RandNet as well. Following their original paper [4], and different from [35], we set up MNIST-4out dataset to contain digit '4' images this time as the outliers and the rest nine classes as inliers. GANomaly performances across 162 HP configs are shown in Fig. 1 (right), where the reported result (red line) in [4] of AUROC $0.795$ appears notably higher than the mean $0.668 \pm 0.105$ that we obtain. (See Appx. A.3 Fig. 9 for all 3 runs.)

For RandNet, we first find and verify a third-party implementation, as the authors could not publicly share theirs, by replicating similar performances to those reported in [11] using the author-recommended HP settings on all 8 datasets. Then for Cardio and Thyroid datasets, we train 192 RandNet ensembles with varying HPs under Polluted as in the original paper. Results from one run are shown in Fig. 1 (right). (See Appx. A.3 Fig. 10 for all 3 runs.) On Cardio, AUROC mean is $0.894 \pm 0.045$ vs. $0.929$ (reported), and Thyroid mean is $0.822 \pm 0.026$ vs. $0.904$ (reported).

In summary, the take-aways from our analysis are as follows. First, it is clear that deep OD models are sensitive to their HP settings, showing that model/HP selection is inevitable. The mean/expected value (of random choice) can be quite away from the best model, motivating this line of research. Second, the recommended settings in recent deep OD papers are arguably optimistic; otherwise more transparency into their selection mechanism is warranted. Finally, hyper-ensemble performance is superior to the mean, possibly owing to different HPs implicitly imposing diversity among constituent models which helps improve detection. Besides performance improvement, hyper-ensembling obviates model selection by joining all models rather than choosing one and is not much sensitive to initialization – setting the stage for our proposed HP-robust OD method ROBOD.

## 4  ROBOD: A Deep Hyper-ensemble for Hyperparameter-Robust OD

### 4.1  Motivation and Overview

The main research question we consider is **RQ1) how to design an unsupervised deep OD model that is robust to its HPs**, i.e. an OD method that has stable, low-variance predictions under varying HPs. Motivated by our sensitivity analysis in Sec. 3, we propose a deep autoencoder (AE) hyper-ensemble model that combines scores from AE models with different HP configurations.

**Definition 4.1 (Hyper-ensemble)** *Given a model family $\mathcal{M}$ with $H$ HPs, let $\boldsymbol{\lambda} \in \mathbb{R}^H$ denote a specific setting of the HPs. A hyper-ensemble averages the output (outlier scores) from a finite number of base models with $m$ different config.s, i.e. $\frac{1}{m} \sum_{i=1}^m \mathcal{M}(\mathbf{x}; \boldsymbol{\lambda}_i)$ for point $\mathbf{x}$.*

Hyper-ensembles exhibit multiple advantages. First, and to our end goal, they ease the model/HP selection burden, bypassing the "choice paralysis". They are less sensitive to random initializations,

---

[3]Besides fixed values for various HPs, Ruff *et al.* [35] recommend HPs that *differ by dataset*; e.g. on MNIST they use 2 CNN modules w/ size 8 and 4 filters, on CIFAR10 they use 3 modules w/ size 32, 64, and 128 filters.

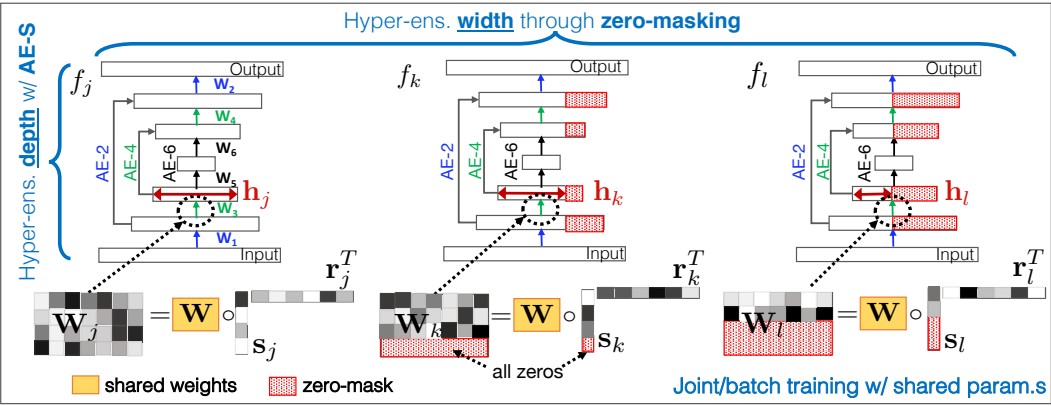

Figure 2: Main design elements in ROBOD: **(1) AE with Skip links (AE-S)**: Each AE-S, denoted $f$, hyper-ensembles multiple AE models with various depths (in the figure, AE-2, 4 and 6), with *shared parameters*; e.g. for $f_j$, layer 2 weights $\boldsymbol{W}_j$ (in dashed circle) is shared among different-depth AEs (i.e. AE-4 and AE-6). **(2) Batch ensemble (BE) training with zero-masks**: Multiple AE-S models with *shared parameters* are trained simultaneously, having various widths thanks to zero-masking; e.g. at layer 2, $\boldsymbol{W}$ is shared by all three different-width AE-S models in the batch (denoted $f_j$, $f_k$, $f_l$), where hidden sizes follow $|\boldsymbol{h}_j| > |\boldsymbol{h}_k| > |\boldsymbol{h}_l|$ through varying size zero-masks on $\boldsymbol{s}_k$ and $\boldsymbol{s}_l$. ($\boldsymbol{sr}^T$ depicts an outer product, and $\circ$ denotes Hadamard/element-wise matrix multiplication.)

with lower variance in performance. Moreover, they can even boost detection performance thanks to the diversity offered by different HPs and ensemble prediction.

The caveat is that deep ensembles are computationally expensive to train. As such, the second research task we tackle is **RQ2) how to speed up hyper-ensemble training**. To that end, we propose novel architectural and training strategies. In a nutshell, these strategies involve three main ideas: (1) we design a multi-layer AE architecture with *skip* connections, denoted AE-S, that helps hyper-ensemble, under a *single* model, *varying-depth* AEs with *shared* parameters; (2) we employ batch ensemble training [42] of multiple AE-S models using *varying size zero-masking* that helps hyper-ensemble *varying-width* AEs all trained *simultaneously* with *shared* parameters; and (3) we train each AE-S on a *subsample* and use out-of-sample scoring. In effect, these strategies allow us to build fewer models, with fewer total number of parameters, on less training data—taking only 2-10% of the time that the naïve ensemble training would take where each model is trained independently.

Fig. 2 illustrates the main design elements of ROBOD pictorially. We present details of the architecture and the training as follows.

## 4.2 Design Strategies for Speeding up Ensemble Training

### 4.2.1 Hyper-ensembling depth: One model for multiple depths

Skip or shortcut connections are applied in NNs for various purposes; e.g., in ResNet [21] for solving the depth degradation problem, in JK-Net [45] for representation learning with varying localities, in U-Net [34] and stochastic depth NNs [24] for training models with adaptive depth, and so on.

Due to the "hourglass" structure of the AE, skipping one layer in an AE would cause dimensionality mismatch for the next layer. Instead, we create long shortcuts by skipping the middle layers while keeping the outer encoder-decoder pairs symmetrically, and refer to this architecture as AE-S.

An illustration of hyper-ensembling depth can be found in Fig. 2; see e.g. the AE-S denoted by $f_j$. Given a $2L$-layer AE-S ($L$ encoder and $L$ decoder layers), there are $2L$ trainable weight matrices. For input $\mathcal{D}$, we generate $L$ outputs by allowing the input to pass through the outermost encoder-decoder pair (which we denote as $AE\text{-}2$), the two outermost encoder-decoder pairs ($AE\text{-}4$), and so on, until the full structure ($AE\text{-}2L$) is traversed by the input. In effect, this trains $L$ different AEs under *one* AE-S model. Then, the overall loss $\mathcal{L}_{\mathsf{AE\text{-}S}}$ of the depth-hyper-ensembling AE-S is the summation of the member AEs' reconstruction errors (denoted $\Delta Err$). Eq. (1) gives an example of the loss function for a 6-layer AE-S as shown in Fig. 2.

$$\mathcal{L}_{\mathsf{AE\text{-}S}} = \Delta Err_{AE\text{-}2}(\mathcal{D}; [\mathbf{W}_1{:}\mathbf{W}_2]) + \Delta Err_{AE\text{-}4}(\mathcal{D}; [\mathbf{W}_1{:}\mathbf{W}_4]) + \Delta Err_{AE\text{-}6}(\mathcal{D}; [\mathbf{W}_1{:}\mathbf{W}_6]) \quad (1)$$

AE-S is an implicit hyper-ensemble which allows simultaneous/joint training of AEs with various depths. It is computationally efficient thanks to parameter sharing, as the outer layers' weights are

reused and tuned among different members. Each ensemble member can also play a regularization effect and prevent an AE from producing low scores for outliers due to overfitting. (See Appx. A.4).

### 4.2.2 Hyper-ensembling width: Zero-masked joint training

BatchEnsemble [42] (BE) is a state-of-the-art parameter-efficient deep ensemble training approach. However, off-the-shelf, it does not allow for *hyper*-ensebling, that is, all models in the batch are trained with the same HP configuration, and share equal size parameters. Our approach builds on BE, adapting it to simultaneously train varying-width AE-S with shared parameters.

First, we briefly review the BE architecture with $K$ AEs. For a specific layer in BE, weight $\mathbf{W} \in \mathbb{R}^{m \times r}$ is shared across all $K$ individual members, while each member $i$ maintains two trainable rank-1 vectors: $\mathbf{s}_i \in \mathbb{R}^m$ and $\mathbf{r}_i \in \mathbb{R}^r$. The outer product $(\cdot)$ of $\mathbf{r}_i$ and $\mathbf{s}_i^T$ creates a "mask" (a matrix in $\mathbb{R}^{m \times r}$) onto the shared weight $\mathbf{W}$ and generates individual weight $\overline{\mathbf{W}}_i$. For the input mini-batch $\mathbf{X}_i \in \mathbb{R}^{n \times m}$, the forward propagation computation is

$$\Phi\big(\mathbf{X}_i \overline{\mathbf{W}}_i\big) = \Phi\big(\mathbf{X}_i\,(\mathbf{W} \circ (\mathbf{s}_i \cdot \mathbf{r}_i^T))\big) = \Phi\big((\mathbf{X}_i \circ \mathbf{s}_i)\mathbf{W} \circ \mathbf{r}_i\big)\,, \tag{2}$$

where $\Phi$ is the activation function, $\circ$ is the element-wise product, and $\mathbf{r}_i$ and $\mathbf{s}_i$ are broadcasted row-wise or column-wise depending on the shapes at play. To vectorize the calculation for all $K$ members in the ensemble, two matrices $\mathbf{S} \in \mathbb{R}^{K \times m}$ and $\mathbf{R} \in \mathbb{R}^{K \times r}$ are constructed with rows of $\mathbf{s}_i$ and $\mathbf{r}_i$, respectively. Thus, for the input $\mathbf{X} \in \mathbb{R}^{Km \times n}$, the BE layer computes the next layer as

$$\Phi\big((\mathbf{X} \circ \mathbf{S})\mathbf{W} \circ \mathbf{R}\big)\,. \tag{3}$$

Since $\mathbf{X}$ is composed of $K$ mini-batching $\mathbf{X}_i$'s tiling up, each member can utilize a different mini-batch of input. The $K$ members are training simultaneously in parallel using one forward pass, thus relieving the memory and computational burden of traditional ensemble methods.

While BE itself is not a hyper-ensemble, we can leverage this architecture to aggregate networks of different widths, also in an efficient way. The widths of a neural network layer correspond to the rows in the weight matrix. For a specific layer, instead of directly operating on $\mathbf{W}$, we instead initialize a zero-one masking vector $\boldsymbol{\alpha}_i \in \{0, 1\}^r$. $\boldsymbol{\alpha}_i$ is to element-wise multiply with the $\mathbf{r}_i \in \mathbb{R}^r$, such that the operation $\mathbf{W} \circ \mathbf{s}_i \cdot (\mathbf{r}_i^T \circ \boldsymbol{\alpha}_i^T)$ will create a masking matrix of size $(m \times r)$ and sparsify the individual weight $\overline{\mathbf{W}}_i$ by allowing zero'ed-out rows. In vectorized notation, the matrix $\mathbf{A} \in \{0, 1\}^{K \times r}$ is composed of $K$ distinct masking vectors. Then, the zero-masked BE forward propagation becomes

$$\Phi\big(\mathbf{X} \circ \mathbf{S}\big)\mathbf{W} \circ (\mathbf{R} \circ \mathbf{A})\big)\,. \tag{4}$$

Sparsifying neural networks has been shown to decrease storage and improve training efficiency, with many algorithms built to wisely prune the neural network [22]. While zero-masked BE is similar to creating neural networks of varying density, our main goal is to obtain models with different widths (i.e. hidden sizes), such that when trained under BE, creates a width-hyper-ensemble. Specifically, we construct the zero-masked BE layer with the maximum width and specify the zeros in respective $\boldsymbol{\alpha}$ vectors, such that the masked-out individual weights correspond to varying-width models.

The zero-masked BE is efficient as $\mathbf{A}$ is fixed throughout training. The element-wise product of the masking incurs little extra time during forward and backward propagation, while all the (rank-1) vectors are cheap to store compared to separate weight matrices as in traditional ensemble training.

### 4.2.3 ROBOD: The overall hyper-ensemble

Let $H$ denote the total number of HPs for an AE-S, where we define a grid of values for each HP; e.g. `depth` $=[4, 6, 8]$, `width` $=[64, 128, 256]$, `lrn_rate` $=[1e^{-3}, 5e^{-4}]$, `drop_out` $=[0.0, 0.2]$, etc.

For the two HPs, `depth` and `width`, we specify the AE-S with the largest depth value (following Sec. 4.2.1, say $2L$) in the grid and leverage the skip connections to obtain the smaller-depth AEs. Similarly, we specify each AE-S in the zero-masked BE with the largest width value (following Sec. 4.2.2, say $K$) in the grid and leverage the zero-masking to obtain other, smaller-width AEs.

Then, a zero-masked BE trains in parallel $K$ varying-width AE-S models, each being an ensemble of $L$ varying-depth AEs. Outlier score for a point $\mathbf{x}$ is averaged across all $KL$ AE models as

$$s(\mathbf{x}) = \frac{1}{KL} \sum_{i=1}^{K} s_i(\mathbf{x})\,, \quad \text{where } s_i(\mathbf{x}) = \sum_{d=1}^{L} \|\mathbf{x} - f_i^{(AE\text{-}2d)}(\mathbf{x}; \boldsymbol{\lambda})\|^2\,, \tag{5}$$

where $f_i$ is the $i$'th AE-S member, $f_i^{(AE\text{-}2d)}$ denotes the AE associated with depth $2d$ within $f_i$, and $\boldsymbol{\lambda}$ is a vector depicting a specific configuration of all the remaining $(H-2)$ HPs; e.g. [lrn_rate $=1e^{-3}$, drop_out $=0.2$, etc.]. Denoting the total number such configurations by $B$, ROBOD averages scores, i.e. the $s(\mathbf{x})$ values, from $B$ different zero-masked BEs as the final outlier score of $\mathbf{x}$.

### 4.2.4 Further speed up by subsampling

As shown in Eq. 2, BE allows mini-batching, where each data point can be used by one or several different ensemble members. This makes BE a natural fit to subsampling, which further expedites the training procedure. To this end, we create $\{\mathbf{X}_i^{in}, \mathbf{X}_i^{out}\}_{i=1}^K$ splits of the training data, where for each AE-S member $i$, we divide the training data into $\mathbf{X}_i^{in}$ and $\mathbf{X}_i^{out} = \mathbf{X}_i/\mathbf{X}_i^{in}$ and solely train on mini-batches from $\mathbf{X}_i^{in}$. We then compute the out-of-sample[4] outlier score of point $\mathbf{x}$ as $s(\mathbf{x}) = \frac{1}{K'L}\sum_{i=1}^K \mathbb{1}(\mathbf{x} \in \mathbf{X}_i^{out})s_i(\mathbf{x})$, where $K' = \sum_{i=1}^K \mathbb{1}(\mathbf{x} \in \mathbf{X}_i^{out})$.

## 5 Experiments

### 5.1 Experimental Setup

**Baselines.** We compare to SOTA *deep* OD methods VanillaAE, RDA [48], DeepSVDD [35] and RandNet [11], the last of which is a deep AE ensemble. We also include the tree-ensemble Isolation Forest [28] (IF) which stands as the SOTA among *traditional* detectors [12]. Besides ROBOD without subsampling, we experiment with two subsampling versions, denoted ROBOD-$\delta$, for sampling rates $\delta \in \{0.1, 0.5\}$. We also compare to the naïve ROBOD with independent training, denoted i-ROBOD.

**Configurations.** The baselines exhibit 2-8 HPs, per which we define a small grid of values (See Appx. A.5 Table 10). We report the expected AUROC performance, i.e., averaged across all configurations in the grid, along with the standard deviation. For ROBOD (and variants) we set $L=6$, and $K=8$; the other HP config.s are listed in Appx. Table 11. To measure sensitivity to random initialization, we average performance across 3 runs. All models are trained on a NVIDIA RTX A6000 GPUs server.

**Datasets.** We conduct experiments on 5 image datasets from MNIST and CIFAR10, as well as 3 tabular datasets from the ODDS repository.[5] MNIST and CIFAR10 are multi-class, where we pick one class as the inliers and subsample the rest at 10% each to constitute outliers. Datasets have varying outlier % and size, with images having high dimensionality. (See Table 3; MNIST-4, -5, -8 depict respective digits as inliers. CIFAR10-0 and -1 refer to airplane and auto class as inliers, respectively.) Details on dataset description and preparation can be found in Appx. A.5.2. The experiments are all conducted under the Polluted (i.e., transductive) setting.

Table 3: Dataset statistics.

| Name | # pts. | dim. | outl.% |
|---|---|---|---|
| MNIST-4 | 6426 | $1\times28\times28$ | 10.0 |
| MNIST-5 | 6426 | $1\times28\times28$ | 10.0 |
| MNIST-8 | 6426 | $1\times28\times28$ | 10.0 |
| CIFAR10-0 | 5500 | $3\times32\times32$ | 10.0 |
| CIFAR10-1 | 5500 | $3\times32\times32$ | 10.0 |
| Thyroid | 3772 | 6 | 2.5 |
| Cardio | 1831 | 21 | 9.6 |
| Lympho | 148 | 18 | 4.1 |

### 5.2 Results

With main results shown in Table 4, we want to answer the following questions: **Q1. How does ROBOD compare to state-of-the-art (SOTA) OD methods?** ROBOD achieves superior performance to all deep OD baselines on MNIST and tabular datasets, and competitive performance on CIFAR10 datasets against the overall runner-up RandNet. Notably, ROBOD performs similarly or even better than i-ROBOD, the latter potentially owing to the guarding effect of parameter sharing against overfitting. Moreover, similar performance can be retained under subsampling, where we train each hyper-ensemble member with 50% or even 10% of the data.

**Q2. How much does ROBOD's performance vary by initialization?** The standard deviation (stdev) of the deep OD baselines is notably large; it is slightly smaller for the ensemble model RandNet, whereas ROBOD has significantly smaller stdev, with near-zero sensitivity to random initialization. The sensitivity of deep baselines suggests that with an arbitrary choice of HPs in the absence of any other guidance, one may acquire much less satisfactory outcomes than the average AUROC.

---

[4]Applicable to transductive OD only; for inductive OD, a point is scored by all ensemble members.
[5]http://odds.cs.stonybrook.edu/

Table 4: AUROC (%) performance of OD methods. Baselines (top) avg.'ed across HP config.s, ROBOD and variants (bottom) avg.'ed over 3 runs w/ random init.s.; ± one stdev. Entries in green depict the method w/ least variance. Highlighted in bold and underline are the **best** and runner-up.

|          | MNIST-4    | MNIST-5    | MNIST-8    | CIFAR10-air | CIFAR10-auto | Cardio     | Thyroid    | Lympho      |
|----------|------------|------------|------------|-------------|--------------|------------|------------|-------------|
| VanillaAE | 81.4±9.4  | 73.6±10.1  | 83.3±4.6   | 60.3±2.0    | 59.2±4.7     | 87.1±7.7   | 81.1 ±8.5  | 89.4±11.7   |
| RDA       | 79.7±11.2 | 68.9±11.4  | 82.6±10.2  | 53.9±8.8    | 54.1±8.9     | 78.4±12.1  | 80.9±5.3   | 78.0±12.2   |
| DeepSVDD  | 81.9±4.3  | 75.7±3.8   | 85.3±3.9   | 55.6±3.1    | 59.0±3.0     | 54.6±7.9   | 67.5±15.1  | 66.5±15.6   |
| RandNet   | 85.3±3.1  | 79.3±3.8   | 85.4±2.2   | 59.1±2.6    | 59.5±3.8     | 89.2±5.3   | 81.5±5.8   | 92.3±8.1    |
| IF        | 84.2±0.8  | 70.1±1.5   | 70.5±1.2   | 42.9±0.5    | **62.7±0.8** | **94.1±0.9** | **97.9±0.4** | **99.5±0.3** |
| ROBOD     | **88.0±0.0** | **81.4±0.1** | 87.8±0.0 | 59.4±0.0    | 59.2±0.1     | 93.5±0.1 | 86.1±0.6 | 98.7±0.1 |
| ROBOD-0.5 | 86.7±0.1  | 78.5±0.1   | 88.1±0.1   | 59.6±0.5    | 58.4±0.3     | 92.1±0.3   | 87.9±1.3   | 98.7±0.1    |
| ROBOD-0.1 | 86.7±0.0 | 78.5±0.1 | **88.2±0.1** | 59.3±0.0 | 58.3±0.2     | 91.8±0.3   | 88.4±0.5   | 99.0±0.1    |
| i-ROBOD   | 84.5±0.0  | 74.6±0.1   | 87.0±0.0   | **62.5±0.0** | 61.7±0.1 | 87.1±0.1 | 93.0±0.0 | 98.9±0.0 |

An interesting observation is regarding the traditional IF baseline, which excels on (low-dim.) tabular datasets, and remains competitive on MNIST-4 and CIFAR10-auto. It also shows relatively small variance w.r.t. its (two) HPs. However, it is significantly inferior on other high-dim. image OD tasks, which may be attributed to its lack of representation learning. Nevertheless, the competitiveness of this simple baseline suggests that traditional OD methods cannot be ignored in the 'horse-race' of developing new deep OD methods, not only for their competitiveness but also for their robustness to only-a-few HPs that makes them easy to employ by practitioners.

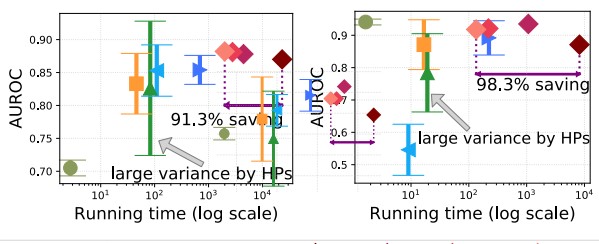

Figure 3: Running time (in log scale) vs. AUROC performance of OD methods (symbols) on (left) MNIST-8 and (right) Cardio. Vertical bars depict one (1) stdev across HP config.s for the baselines.

In Fig. 3 we show the running time (in sec.s) versus the performance for all OD methods (colored symbols) on one image and one tabular dataset for brevity. (See Appx. A.6 Fig. 12 for remaining datasets.) ROBOD achieves higher performance, with lower, near-zero variability compared to deep baselines. Traditional IF, based on randomized trees, is very fast, yet it often underperforms on high dimensional tasks such as with images.

**Q3. What are the savings in running time compared to naïve hyper-ensemble w/ independent training?**

We study the savings that ROBOD offers compared to the naïve hyper-ensembling. As shown in Table 5, it runs 3-10× faster across datasets. With subsampling at 10%, ROBOD-0.1 runs in about 1.6-10% of what it takes by i-ROBOD. Fig. 4 (on MNIST-5) shows that while i-ROBOD time increases by the number of models, i.e. larger $L$ and $K$ (fixing other HPs), ROBOD takes near-constant time thanks to the batch/simultaneous training of varying-depth and -width members.

Table 5: Running time (sec.) of the naïve ensemble i-ROBOD vs. ROBOD. ROBOD-0.1 that trains with 10% subsampling offers 90-98% relative savings in training time.

|           | MT-4  | MT-5  | MT-8  | CIF-air | CIF-auto | Cardio | Thy.  | Lym.  |
|-----------|-------|-------|-------|---------|----------|--------|-------|-------|
| i-ROBOD   | 24192 | 24656 | 23165 | 38830   | 39562    | 8205   | 14313 | 1804  |
| ROBOD     | 4521  | 4630  | 4456  | 10721   | 10722    | 1070   | 2049  | 180   |
| ROBOD-0.1 | 2241  | 2141  | 1997  | 2834    | 3003     | 134    | 295   | 44    |
| Savings (%) | 90.74 | 91.32 | 91.38 | 92.70 | 92.41    | 98.37  | 97.94 | 97.56 |

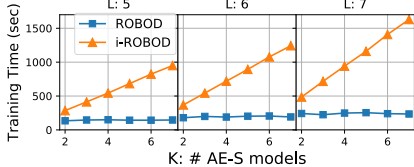

Figure 4: Runtime comparison of ROBOD to i-ROBOD w/ varying $L$ and $K$.

**Q4. How sensitive hyper-ensembles are to their own HPs, which include (1) the selected HP ranges, and (2) the number of sub-models?** The two HP terms are directly related, since expanding/shrinking the HP ranges result in more/fewer sub-models constituting the ensemble. We first answer how HP value ranges affect the accuracy of hyper-ensembles. With MNIST-4, we expand, shrink or shift the HP ranges for the VanillaAE, which are consitituents of i-ROBOD. The experimental details are provided in Appx. A.7.1. As shown in Table 6, i-ROBOD achieves more stable results across all settings. Moreover, i-ROBOD produces lower variance with respect to its own HPs than that of the individual model results.

We also study how number of sub-models impact our model. For the same MNIST-4, DeepSVDD and VanillaAE are selected as the base model for hyper-ensembles, with details in Appx. A.7.2. Fig. 5

Table 6: Mean and stdev of AUROCs for i-ROBOD vs. VanillaAE. i-ROBOD's performance is evaluated with different HP ranges in Table 12. Mean and stdev for VanillaAE are calculated over all constituent sub-models from the same table. The hyper-ensemble i-ROBOD has considerably lower performance variation w.r.t. its HPs, than individually traind VanillaAE.

| Mean&Std. (i-ROBOD) | Mean&Std. (VanillaAE) |
|---|---|
| 83.0±**2.8** | 78.9±9.8 |

shows the AUROC corresponding to different number of sub-models (for DeepSVDD we have both Polluted (left) and Clean (middle) settings, and for VanillaAE we have Polluted (right) setting only). Observe that AUROC quickly stabilizes and the variance shrinks, as the number of sub-models grows beyond a certain size. The larger number of sub-models is, the more stable is the performance.

Finally, we conduct the sensitivity analysis to both HP value ranges and the number of sub-models (since they are correlated) for our proposed ROBOD. We evaluate on the Cardio dataset and provide the details in Appx. A.7.3. Table 7 shows that ROBOD yields smaller variance to its own HPs and provides more competitive results than other benchmarked models. To summarize, our experiments show that larger number of sub-models and expanded HP ranges under fine grids can relax an hyper-ensemble model's dependency on its own HPs.

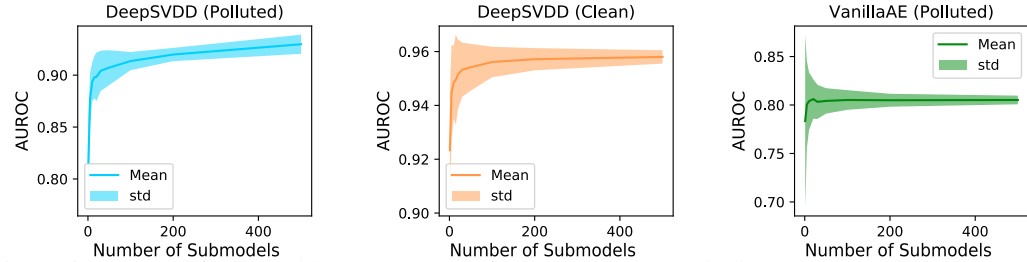

Figure 5: Number of sub-models vs. AUROC of ensembled-OD methods on MNIST-4. Shaded areas depict one (1) stdev around the mean performance of the ensemble. For all cases the performance quickly improves to a stable state after ∼20 sub-models, with notably small standard deviation.

Table 7: For Cardio dataset, we provide mean and stdev AUROCs of ROBOD and benchmarks from HP ranges in Tables 15 (ROBOD) and 10 (benchmarks).The hyper-ensemble ROBOD has considerably lower performance variation w.r.t. its HPs.

| ROBOD | VanillaAE | RDA | DeepSVDD | RandNet | IF |
|---|---|---|---|---|---|
| 93.6±**0.4** | 87.7±7.7 | 78.4±12.1 | 54.6±7.9 | 89.2±5.3 | 94.1±0.9 |

# 6 Conclusion

In this work, we have provided a thorough analysis on the hyperparameter (HP) sensitivitiy of several state-of-the-art deep outlier detection (OD) methods. Our findings quantitatively confirm that model selection is vital and advocate efforts in this line of research. To this end, we introduce ROBOD, a scalable hyper-ensemble OD method that remedies the "choice paralysis" by assembling various autoencoder models of different HP configurations, hence obviating HP/model selection. We speed up ensemble training through novel strategies that simultaneously train varying depth and width models under parameter sharing. Extensive experiments on image and point-cloud datasets show the competitiveness of ROBOD compared to existing OD baselines, while providing consistent results across random initializations. We hope that our work increases awareness to the unsupervised model selection challenge for the newly booming deep OD literature and motivates future work on hyperparameter-robust model design.

## Acknowledgments

This work is sponsored by NSF CAREER 1452425. We also thank PwC Risk and Regulatory Services Innovation Center at Carnegie Mellon University. Any conclusions expressed in this material are those of the author and do not necessarily reflect the views, expressed or implied, of the funding parties.

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
