# A    Appendix

## A.1    Preview of Existing Deep OD Methods

Table 8: Representative unsupervised deep OD models from 4 different families (for a broader coverage, see surveys [31, 36, 9]), annotated in terms of data used for training, test, and validation/model selection (if any). No existing work attempts (unsupervised) model selection; vast majority reports results for a *fixed* (how, unclear) "recommended" config. or tune *some* but not all HPs using labeled validation and even test (!) data. AE: autoencoder, SSL: self-supervised learning, Clean: inlier-only.

| Method | Year | Family | Train | Test | Validation (HP/Model Selection) |
|--------|------|--------|-------|------|---------------------------------|
| RandNet [11] | 2017 | AE ensemble | Polluted | =Train | None, fixed – sensitivity analysis on some HPs |
| RDA [48] | 2017 | AE | Polluted | =Train | Best $\lambda$ on Test, other HPs fixed |
| DAGMM [49] | 2018 | AE & density | Clean & Pol.d | Disjoint | None, fixed – sensitivity on reg. param.s $\{\lambda_1, \lambda_2\}$ |
| DeepSVDD [35] | 2018 | One-Class | Clean | Disjoint | Best $\nu$ on Test, other HPs fixed |
| DROCC [18] | 2020 | One-Class | Clean | Disjoint | Validation data to tune some (not all) HPs |
| HRN [23] | 2020 | One-Class | Clean | Disjoint | 10% of Test for tuning $\{\lambda, n\}$, other HPs fixed |
| (f-)AnoGAN [37, 38] | 2017 | GAN | Clean & Pol.d | Disjoint | None, fixed |
| EGBAD [46] | 2018 | (Bi)GAN | Clean | Disjoint | None, fixed |
| GANomaly [4] | 2018 | GAN | Clean | Disjoint | Best reg. weights $\{w_{adv}, w_{con}, w_{enc}\}$ on Test, others fixed |
| GOAD [5] | 2020 | SSL | Clean | Disjoint | None, fixed |
| NeuTraL [6] | 2020 | SSL | Clean | Disjoint | 10% of Test for tuning transformation HPs, others fixed |

## A.2    Details on Hyperparameter-Sensitivity Analysis

**Clean versus Polluted Testbed Setup.** For sensitivity analysis, we construct our testbed on several datasets, including MNIST, CIFAR10, Thyroid and Cardio. For MNIST, we choose Digit '4' and '5' as the inlier-class, individually. For CIFAR10, we choose class 'automobile' as the inlier-class. The inlier-class is assigned the label 0, while we regard all classes other than the inlier-class as outliers and mark them with label 1 instead. Since MNIST and CIFAR10 are image data, we first apply the global contrast normalization to each individual image. We utilize the default train/test data-split (supported with Pytorch vision package). In the Clean setting, we select only the inlier-class data from train data-split as the training data. We measure and report the AUROC of the compared methods on the test data-split, with label 0 being the inlier-class and 1 being rest of the classes. In the Polluted setting, we utilize all the inlier-class data from the train data-split as the label 0, and we mix the data with 10% outlier-classes' data within the train data-split.

The tabular data, Thyroid and Cardio, are downloaded from the ODDS repository (available at http://odds.cs.stonybrook.edu/. The data come in Polluted setting, with 2.5% and 9.6% outliers, respectively. The inliers are labeled 0 and outliers are denoted as label 1. For tabular data, we transform and scale each feature between zero and one.

We also conduct experiment using GANomaly [4]'s data (MNIST digit '4' as the outlier class, rest as inliers). The data split and configuration are the same as described in the authors' provided code.

**Model HP Descriptions and Grid of Values.**

- VanillaAE:
    1. `n_layers`: number of encoder layers (i.e. depth)
    2. `layer_decay`: the rate of NN width's shrinkage between current and next encoder layers, the decoder layers are expanded at the same rate.
    3. `LR`: learning rate
    4. `iter`: number of epochs/iterations
- RDA:
    1. $\lambda$ (model-specific reg.): a penalty term that tunes the level of sparsity in the outlier matrix $S$ (refer to Section 3.1 in [48]).
    2. `n_layers`: number of encoder layers (i.e. depth)
    3. `layer_decay`: the rate of NN width's shrinkage between current and next encoder layers, the decoder layers are expanded at the same rate.
    4. `LR`: learning rate
    5. `inner_iter`: number of epochs/iterations to train the underlying autoencoder (AE), before updating the outlier matrix $S$ and inlier matrix $L$ (refer to Section 4.1 in [48]).
    6. `iter`: number of epochs/iterations in the algorithm, which first separates the training data into outlier matrix $S$ and inlier matrix $L$, then trains AE on the inlier matrix.

Table 9: We define a grid of 1-3 unique values for each hyperparameter (HP) of each deep OD method studied. With 4-to-8 different HPs each, the total number of configurations, and i.e. models trained, quickly grows to several hundreds. When applicable/available, we include the **author-recommended** value (marked in bold and underlined) in the respective grid.

| Method | Hyperparameter | Grid | #values | Method | Hyperparameter | Grid | #values |
|--------|----------------|------|---------|--------|----------------|------|---------|
| VanillaAE | n_layers | [2, 3, 4] | 3 | DeepSVDD | conv_dim | [**8**, 16, 32] | 3 |
| | layer_decay | [1, 2, 4] | 3 | | fc_dim | [**16**, 32] | 2 |
| | LR | [1e-3, 1e-4, 1e-5] | 3 | | Relu_slope | [**1e-1**, 1e-3] | 2 |
| | iter | [200, 500, 1000] | 3 | | pretr_iter | [200, **350**, 400] | 3 |
| | | Total = | 81 | | pretr_LR | [1e-4, 1e-5] | 2 |
| RDA | $\lambda$ | [5e-1, 5e-3, 5e-5] | 3 | | iter | [100, 200, **250**] | 3 |
| | n_layers | [2, 3, 4] | 3 | | LR | [1e-4, 1e-5] | 2 |
| | layer_decay | [1, 2, 4] | 3 | | wght_dc | [1e-5, **1e-6**] | 2 |
| | LR | [1e-3, 1e-4] | 2 | | | Total #models = | 864 |
| | inner_iter | [20, 50] | 2 | RandNet | n_layers | [3, 5, **7**, 9] | 4 |
| | iter | [5, 20, 50] | 3 | | layer_decay | [0.3, **0.6**] | 2 |
| | | Total #models = | 324 | | sample_r | [1.00, **1.01**] | 2 |
| GANomaly | $w_{adv}$ | **1** | 1 | | ens_size | [**50**, 200] | 2 |
| | $w_{con}$ | [25, **50**, 100] | 3 | | pretr_iter | 100 | 1 |
| | $w_{enc}$ | [0.1, **1**] | 2 | | iter | [**300**, 1000] | 2 |
| | z_dim | [50, **100**, 200] | 3 | | LR | [1e-2, 1e-3, 1e-4] | 3 |
| | LR | [5e-3, **2e-3**, 5e-4] | 3 | | wght_dc | 0 | 1 |
| | iter | [10, **15**, 25] | 3 | | | Total #models = | 192 |
| | | Total #models = | 162 | | | | |

- DeepSVDD:
    1. conv_dim: the output number of channels, after the first- convolutional encoder layer. After the first-layer, the number of channels expand at rate of 2.
    2. fc_dim: the output dimension of the fully connected layer between convolutional encoder layers and decoder layers, in the LeNet structure [27].
    3. Relu_slope: DeepSVDD utilizes leaky-relu activation to avoid the trivial, uninformative solutions [35]. Here we alter the leakiness of the relu sloping.
    4. pretr_iter: In [35], an AE is pre-trained, to set the hypersphere center $c$ to the mean of the mapped data. pretr_iter determines the number of epochs/iterations to train AE.
    5. pretr_LR: the learning rate to pretrain the AE.
    6. iter: the number of epochs/iterations in training the DeepSVDD
    7. LR: the learning rate during training
    8. wght_dc: weight decay rate
- GANomaly
    1. $w_{adv}$: weight parameter that adjusts the adversarial loss function (See Section 3.2 in [4].)
    2. $w_{con}$: weight parameter that adjusts the contextual loss, for learning the contextual information about the input data (See Section 3.2 in [4].)
    3. $w_{enc}$: weight parameter that adjusts the encoder loss and minimizes the distance between the bottleneck features and the encoder of the generated features (See Section 3.2 in [4].)
    4. z_dim: the dimension of the reduced embedded space after the input is passed through the encoders
    5. LR: learning rate
    6. iter: number of epochs/iterations
- RandNet
    1. n_layers: number of encoder layers (i.e. depth) in BAE
    2. layer_decay: the rate of NN width's shrinkage between current and next encoder layers, the decoder layers are expanded at the same rate.
    3. sample_r: sample size selection for adaptive sampling (See Section 3.3 in [3].)
    4. ens_size: number of ensemble members/models
    5. pretr_iter: number of epochs/iterations to pretrain the BAE
    6. iter: number of epochs/iterations
    7. LR: learning rate
    8. wght_dc: weight decay rate

## A.3 Additional Results: Hyperparameter-Sensitivity Analysis

In Fig. 6, we show the AUROC performance of deep OD methods in Clean setting. In Fig. 7, Fig. 8, Fig. 9 and Fig. 10 we show additional experiment results and AUROC performances over 3 runs with different random initializations.

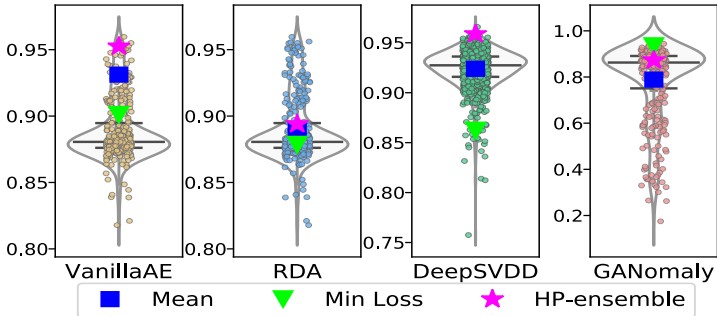

Figure 6: AUROC performance of deep OD methods with different HP configurations (circles) on MNIST-4 with Clean training data showcase notable variation (i.e., sensitivity). Hyper-ensemble (☆) improves notably over Mean (□).

## A.4 Regularization Effect of Weight Sharing

One noticeable problem with applying autoencoders (AE) and reconstruction loss for outlier detection is that for AE, the decision boundary is hard to draw due to noise and outliers that can impact the quality of the reconstruction. In some cases, the AE can overfit to all the data points including the outliers, causing large false negative rate. In other cases, it may underfit to data and fail to reconstruct the input satisfactorily. While the "denoising AEs" [44] or "correntropy AEs" [32] may help to alleviate such problems, they both rely on *clean* inlier-only data, which is typically not available in real-world scenarios. On the other hand, ROBOD with AE-S allows an implicit ensemble of various NN depths and widths and plays a regularization effect on the outliers scores, in effect helping prevent AE from potential failures due to underfitting or overfitting.

To demonstrate the regularization effect with AE-S, we compare the reconstructed images of AE versus AE-S, with number of layers $\{2, 4, 6, 8, 10, 12, 14\}$, respectively (hidden dimension decays at a constant rate of 2). We compare the individual AE and AE-S, keeping the other HP-configurations same, and training under the Polluted setting with MNIST digit '5' as the inlier class. Fig. 11 shows the reconstructed images with these 7 individual AEs and a single AE-S ensemble, with implicit AE-i ($i \in \{2 \ldots 14\}$). We see that individual AEs are overfitting to the outlier classes (digit '3' and '7') providing good reconstructions when layers $L$ equal to 2 and 4. In contrast, they underfit and fail to reconstruct any inliers or outliers when layers $L$ equals to 6, 8, and 10. When $L$ increases to 12 and 14 layers, individual AEs provide low-quality reconstructions to both inliers and outliers, distorting all outliers to inlier class (digit '5'). In contrast, AE-S provides lower-resolution reconstruction for outlier classes during *AE-2* and *AE-4*, that is overfitting showcases at a lower degree. AE-S can still provide signal to distinguish the outliers from the inliers from *AE-6* through *AE-12* where the outliers gradually become more blurred and start to deform into inlier's shape. Only at very large depth at *AE-14*, AE-S cannot distinguish between outlier instances from inlier instances, providing low quality predictions to both classes.

Intuitively, this regularization effect is due to the weight sharing between AE-i's. Since the next AE-i utilizes the weights optimized by the previous AE-i, the training phase becomes easier and underfitting is less likely to occur. Moreover, AE-S has a similar structure as U-Net [34], which is known to reduce the overfitting in medical image segmentation tasks.

While both i-ROBOD and ROBOD average the reconstruction loss from ensemble members, ROBOD with AE-S structure (and hence parameter sharing) is able to better capture the outlier information thanks to this regularization effect. This phenomenon also explains why ROBOD's performance is better than that of i-ROBOD, e.g. on MNIST datasets in Table 4.

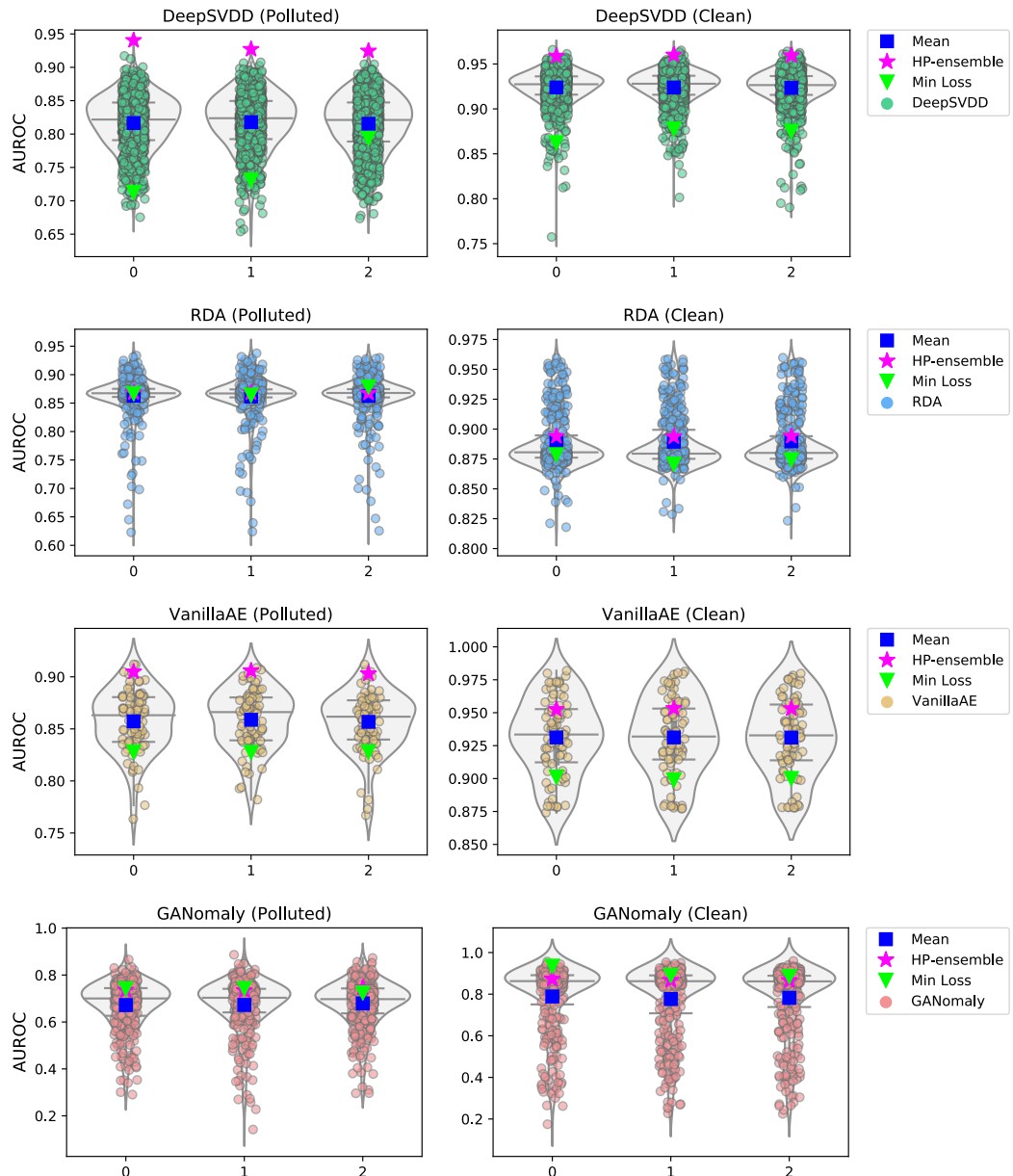

Figure 7: AUROC results under varying HP-configurations on MNIST-4 dataset under the Clean (only train on digit '4' images) and Polluted (digit '4' as inliers, the rest nine classes down-sampled at 10% as outliers) settings. We conduct each experiment 3 times with different random initialization, where each plot's x-axis corresponds to the experiment index. Note: y-axes are not directly comparable – we use different y-axis to better reflect the spread for each experiment.

## A.5 Details on Experiment Setup

### A.5.1 Hyperparameter Configurations: Details

In experiments, we compare to VanillaAE, RDA [48], DeepSVDD [35] and RandNet [11]. We have not compared to GANomaly due to the higher variance of performances we observed during sensitivity analysis. We define a small grid of values for the HPs of each of these methods.

Because DeepSVDD is originally trained with LeNet [27] (Convolutional AE) structure, we also implement Convolutional AEs for algorithms that are either pretrained with AEs, or utilize AEs as the backbone algorithm. The detailed HP configurations are shown in Table 10. The VanillaAE and

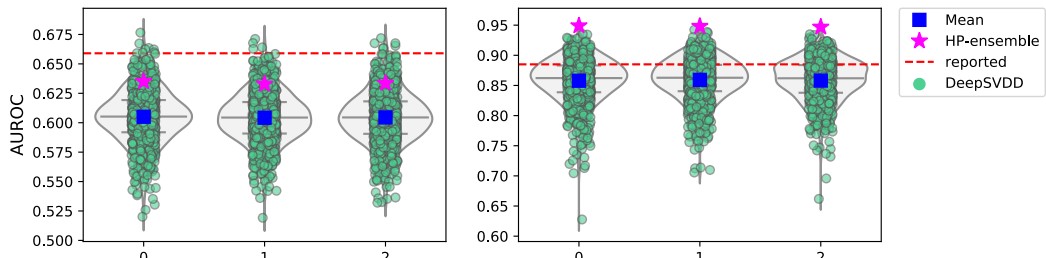

Figure 8: For DeepSVDD algorithm, we show various HP-configurations' AUROC results under the Clean setting. Left: MNIST digit '5' as the inlier data. Right: CIFAR10 'automobile' as the inlier data. We conduct each experiment 3 times with different random initialization, where each plot's x-axis corresponds to the experiment index. Performance reported in the original paper [35] appears above what we have obtained on average for both datasets.

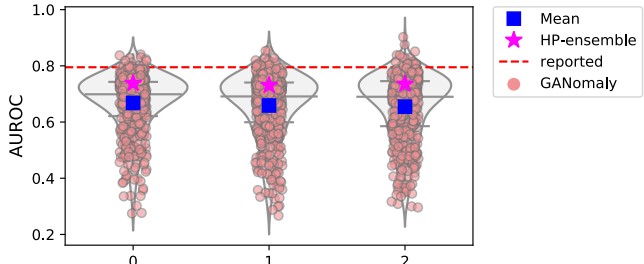

Figure 9: AUROC results with varying HP configurations for the GANomaly algorithm. We utilize the same experimental setting as in [4], with MNIST digit '4' as the outlier class, and rest of the digit classes as inliers (also in Clean setting). The x-axis corresponds to the experiment index among 3 independent runs each with a different random initialization. Performance reported in the original paper [4] appears above what we have obtained on average.

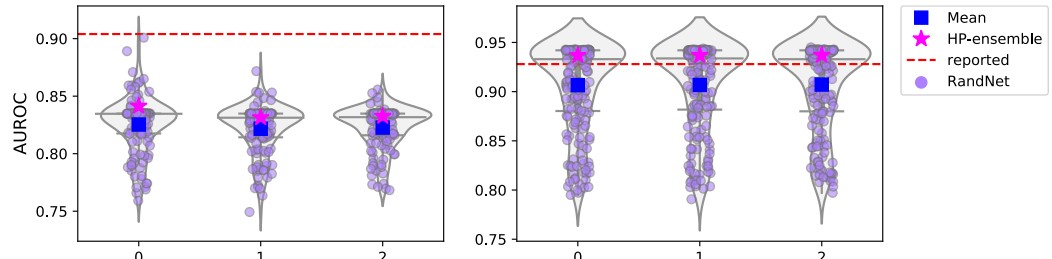

Figure 10: AUROC results with varying HP configurations for the RandNet algorithm. As RandNet implementation contains fully connected layers, we evaluate it only on tabular data under Polluted setting, similar to the experiments in [3]. Left: Thyorid dataset, Right: Cardio dataset. Each plot's x-axis corresponds to the experiment index for 3 different runs with random initialization.

RandNet are trained with AE for all three kinds of datasets; DeepSVDD is trained with Convolutional AE (LeNet) on image data and AE on tabular data; RDA is trained on AE for MNIST and tabular data, it utilizes Convolutional AE (LeNet) on CIFAR10. If an algorithm is trained with AE as the underlying structure, we define a shared grid of HPs: number of encoder layers, decay rate (the rate of NN width's shrinkage between current and next encoder layers), dropout rate, train learning rate, etc. Similarly for Convolutional AE, the algorithms apply the same grid of HPs: convolution channels, fully-connected layer dimensions, weight decay, learning rate, etc.

With respect to model-specific HPs, RDA uses $\lambda$ as a penalty constant for sparsity of the outlier matrix. It also uses `inner_iters` and `iter`, to specify the number of epochs respectively for training an AE and for separating the data into outlier and inlier matrices. For RandNet, we fix the number of ensemble members to 5 due to computational overhead in training high-dimensional image datasets, while [3] uses 50 ensemble members as RandNet is trained on tabular data only. Other model-specific HP descriptions can be found in Sec. A.2 Model HP Descriptions and Grid of Values.

Table 10: Grid of values for the HPs and neural architectures used in experiments.

| HPs | AE | LeNet (MNIST) | LeNet (CIFAR10) |
|---|---|---|---|
| Number of encoder layers | [2,3,4,5,6] | [2] | [3] |
| Decay rate | [1.5,1.75,2,2.25,2.5,2.75,3,3.25] | - | - |
| Convolution channels | - | [8] | [16] |
| FC layer dimensions | - | [16,32,64] | [32,64,128] |
| Dropout rate | [0.0,0.2] | - | - |
| Weight decay | [0,1e-5] | [0,1e-5,1e-6] | [0,1e-5,1e-6] |
| Train Learning Rate | [1e-3,1e-4] | [1e-4,1e-5] | [1e-4,1e-5] |

| NN settings for dataset: | MNIST | CIFAR10 | Tabular Data |
|---|---|---|---|
| VanillaAE | AE | AE | AE |
| DeepSVDD | LeNet (MNIST) | LeNet (CIFAR10) | AE |
| RDA | AE | LeNet (CIFAR10) | AE |
| RandNet | AE | AE | AE |

| Method | Other HP settings |
|---|---|
| VanillaAE | train iters: [250,500] |
| RDA | $\lambda$: [1e-1,1e-3,1e-5], iters: [20,30], inner iters: [20,30] |
| DeepSVDD | LeakyRelu Slope: [1e-1, 1e-3], pretrain iters: [100,350], pretrain lr: [1e-4], train iters: [250,500]] |
| RandNet | pretrain iters: [100], pretrain lr:[1e-4] adaptive sampling rate: [1.0] train iters: [250,500], ens_size = [5] |

For each data point, i-ROBOD provides a score by averaging each individual VanillaAE's reconstruction error for a data point, thus the training time required for i-ROBOD sums up each VanillaAE's time. ROBOD speeds up i-ROBOD with fast ensembles across varying NN depths and widths. Table 11 shows the architecture overview for ROBOD. Specifically, the NN depths and widths are set to 8 and 6, representing the implicit ensembles. ROBOD explicitly ensembles over various train iterations, learning rates, dropout rate, weight decay, providing an averaged reconstruction error score for each data point. We also experiment with two subsampling based versions, denoted ROBOD-$\delta$, where $\delta = 0.1$ and $0.5$, respectively. We let ROBOD-$\delta$ to train each ensemble member only on $10\%$ or $50\%$ of the training data, and score "out-of-sample" points, i.e. the rest of the (unseen) data points.

Table 11: ROBOD architecture overview

| List of HPs | Settings |
|---|---|
| BatchEnsemble num_models | 8 (implicit ensemble over decay rate: [1.5,1.75,2,2.25,2.5,2.75,3,3.25]) |
| num_layers | 6 (implicit ensemble over AE-2, AE-4, AE-6,AE-8,AE-10,AE-12) |
| Train iterations | [250,500] |
| Train Learning Rate | [1e-3,1e-4] |
| Dropout rate | [0.0, 0.2] |
| Weight decay | [0, 1e-5] |

### A.5.2 Dataset Description

Similar to the experiment settings in our sensitivity analysis, we evaluate the baseline methods and ROBOD on image data (MNIST, CIFAR10) as well as tabular data (Thyroid, Cardio and Lympho). For MNIST, we conduct three sets of experiments; each chooses digit '4','5', or '8' as the inlier class, respectively. For CIFAR10, we conduct two sets of experiments, with 'airplane' (CIFAR10-0) and 'automobile' (CIFAR10-1) as the inlier classes. For image data, we employ global contrast normalization to individual images. The inliers are assigned the label 0 and all classes other than the inlier class will be marked with 1, indicating the outlier class. We conduct all experiments under the Polluted setting, where we use all the inlier class points from Pytorch's train data-split as label 0, and combine them with $10\%$ of points from the outlier classes within the train data-split. The tabular data, Thyroid, Cardio and Lympho, are downloaded from the ODDS repository (available at http://odds.cs.stonybrook.edu/, which contain $2.5\%$, $9.6\%$ and $4.1\%$ outliers, respectively. Similar to image data, the inliers have label 0 and outliers have label 1. Prior to model training, we transform and scale each feature between zero and one using the MinMaxScaler.

## A.6 Additional Experiment Results

Fig. 11 is shown to illustrate the regularization effect of parameter sharing in AE-S versus a vanilla AE. (See A.4 for discussion.)

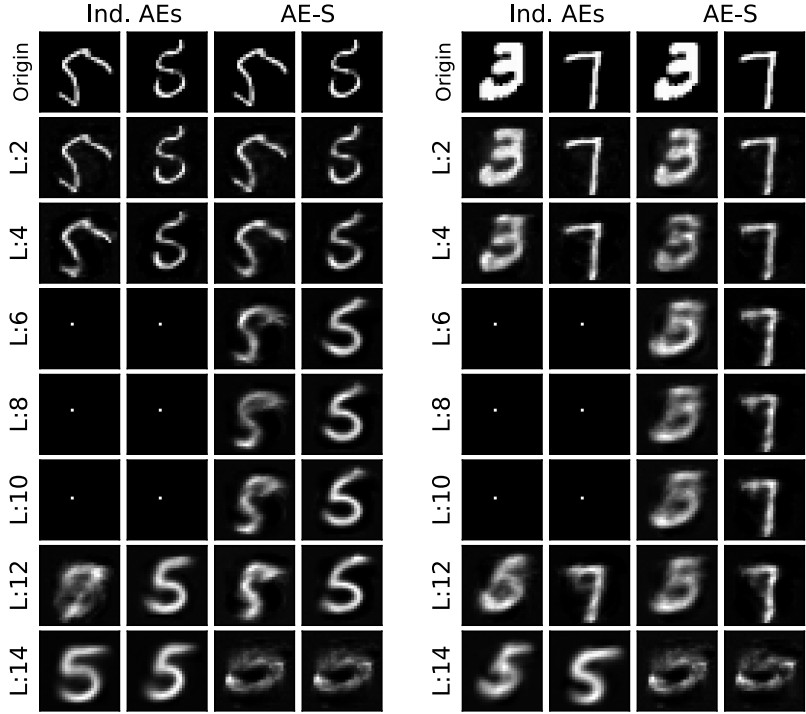

Figure 11: Left: Reconstructed *inlier* class instances (MNIST digit '5'), generated by individual AEs vs AE-S structure in ROBOD. Right: Reconstructed *outlier* class instances (MNIST digit '3' and '7') by individual AEs versus AE-S structure. L denotes the number of layers.

Fig. 12 shows the running time (in log scale) vs. AUROC performance of OD methods (symbols) on datasets MNIST-4, MNIST-5, CIFAR-airplane, CIFAR-automobile, Thyroid and Lympho.

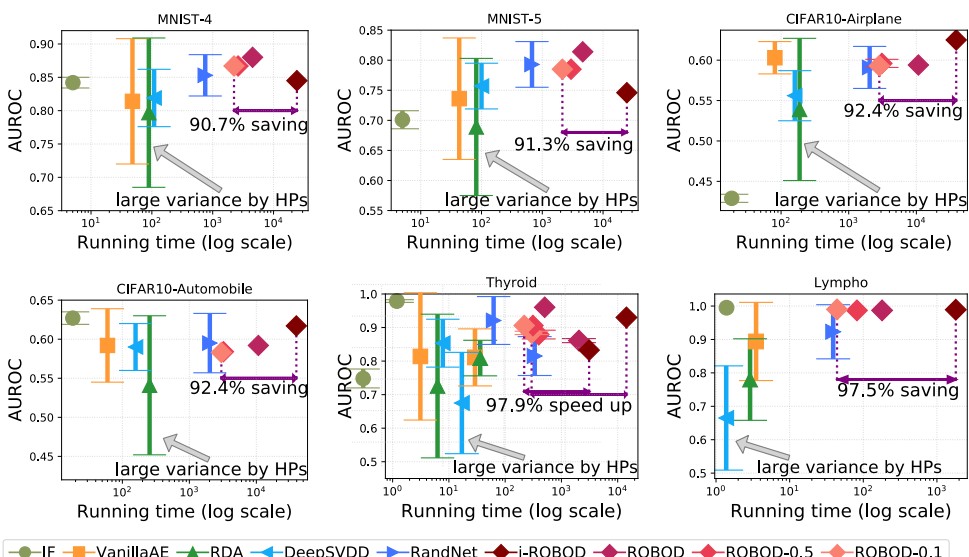

Figure 12: Running time (in log scale) vs. AUROC of OD methods (symbols) on other datasets. Vertical bars depict one (1) stdev across HP config.s. ROBOD often improves detection performance and importantly, provides robust low-variance performance. Sampling based ROBOD reduces running time considerably with small difference in relative performance.

### A.7    HP Sensitivity Analysis for Ensemble Models

In this section, we want to show whether or not hyper-ensembles are robust to their own HPs: (1) the HP value ranges, and (2) the number of sub-models. We remark that the number of sub-models and HP ranges are directly related for a *hyper*-ensemble, since expanding (or shrinking) the HP ranges result in more (or fewer) sub-models constituting the ensemble. Finally, we show how the proposed ROBOD is robust to both the number of sub-models and varying HP value ranges.

### A.7.1    How do HP value ranges affect the ensemble?

Starting with the same HP settings as in Table 10 for VanillaAE, we expand, shrink or shift the HP ranges for the VanillaAE models, and then measure the resulting accuracy and variance of i-ROBOD, which assembles multiple VanillaAEs. Table 12 summarizes the various actions taken as compared to the original settings. For example, in Row 1, the range for the number of auto-encoders is expanded to also include $[7, 8, 9]$ layers, or shifted to exhibit auto-encoders with $[6, 7, 8, 9]$ layers, shifting from original $[2, 3, 4, 5, 6]$ layers. Other HPs are also altered in a similar fashion.

Table 12: Overview of i-ROBOD with several HP ranges

| List of HPs | Original Setting | Actions |
|---|---|---|
| Number of encoder layers | [2,3,4,5,6] | Expand:[2,3,4,5,6,7,8,9], Shift:[6,7,8,9], Shrink:[2,3,4] |
| Train iterations | [250,500] | Shift&Shrink:[1000], Expand:[250,500,1000] |
| Decay rate | [1.5-3.25] | Shrink:[1.5-2.0] |
| Train Learning Rate | [1e-3,1e-4] | Shift:[1e-2,1e-3], Expand:[1e-2,1e-3,1e-4] |
| Dropout rate | [0.0, 0.2] | Shift:[0.2,0.5], Expand:[0.0,0.2,0.5] |
| Weight decay | [0.0, 1e-5] | Shift: [1e-3,1e-4], Expand: [1e-3,1e-4,1e-5,0.0] |

We conduct our experiments on the same MNIST dataset with '4' being the inlier digit. For each experiment, we alter a subset of the HPs from Table 12 and record the AUROC across 3 runs. We measure the mean and standard deviation of i-ROBOD AUROC in comparison with VanillaAE's. Table 6 provides the mean and variance of i-ROBOD, evaluated on the performances over all these experimental runs, in comparison to training individual VanillaAE. (Detailed results are in Table 13.)

Our results shows that i-ROBOD produces more stable results in all settings than individually trained models. Moreover, it produces lower variance with respect to its own HPs (2.8) than that of the individual model results (9.8). Because of the lack of any prior knowledge, for many new models and architectures, finding a "good" HP range (a set of HPs that potentially contain the best HP for the task) can be hard. In that case, both the hyper-ensemble and an individual model may achieve less than satisfactory performance. However, hyper-ensemble is likely to achieve better stability to the range of HPs, while individual models are more sensitive when finding the optimal range is unreachable due to lack of validation.

Table 13: The altered HP ranges and AUROC results for i-ROBOD and VanillaAE. For example, "Number of encoder layers:[6,7,8,9]" means the values are shifted from [2,3,4,5,6] in the original experiment settings to [6,7,8,9], while the other HPs are the same as before as shown in Table 12.

| Altered HP Ranges | Mean&Std. (i-ROBOD) | Mean&Std. (VanillaAE) |
|---|---|---|
| No Changes | 84.5±**0.0** | 81.4±9.4 |
| Number of encoder layer: [2,3,4] | 86.1±**0.0** | 85.1±6.0 |
| Number of encoder layer:[6,7,8,9] | 77.4±**0.1** | 75.1±7.1 |
| Number of encoder layer: [2,3,4,5,6,7,8,9] | 82.1±**0.0** | 79.6±8.9 |
| Train learning rate: [1e-2,1e-3] | 83.3±**0.2** | 79.7±12.5 |
| Train learning rate: [1e-2,1e-3,1e-4] | 83.2±**0.1** | 79.4±12.3 |
| Dropout rate: [0.2,0.5] | 83.6±**0.2** | 80.7±9.5 |
| Dropout rate: [0.0,0.2,0.5] | 84.1±**0.1** | 81.1±9.6 |
| Train iterations:[1000] | 84.5±**0.2** | 80.1±9.6 |
| Train iterations:[250,500,1000] | 84.5±**0.1** | 81.2±9.0 |
| Weight decay: [1e-3,1e-4] | 79.4±**0.1** | 77.9±3.8 |
| Weight decay: [1e-3,1e-4,1e-5,0.0] | 82.6±**0.1** | 80.2±8.2 |
| Decay rate: [1.5,1.75,2.0] | 81.4±**0.1** | 78.6±13.4 |

Table 14: We define a grid of values for each HP of our studied DeepSVDD and VanillaAE. With 4-to-8 different HPs each, the total number of configurations quickly grows to several hundreds.

| Method | Hyperparameter | Grid | #values | Method | Hyperparameter | Grid | #values |
|---|---|---|---|---|---|---|---|
| VanillaAE | n_layers | [2,3,4,5,6,7,8,9] | 8 | DeepSVDD | conv_dim | [8, 16, 32] | 3 |
| | layer_decay | [1.5,1.75,2,2.25,2.5,2.75,3,3.25] | 8 | | fc_dim | [16, 32] | 2 |
| | LR | [1e-2, 1e-3, 1e-4] | 3 | | Relu_slope | [1e-1, 1e-3] | 2 |
| | iter | [200, 500, 1000] | 3 | | pretr_iter | [200, 350, 400] | 3 |
| | wght_dc | [1e-3,1e-4,1e-5,0] | 4 | | pretr_LR | [1e-4, 1e-5] | 2 |
| | Dropout | [0.0, 0.2, 0.5] | 3 | | iter | [100, 200, 250] | 3 |
| | | Total #models = | 6,912 | | LR | [1e-4, 1e-5] | 2 |
| | | | | | wght_dc | [1e-5, 1e-6] | 2 |
| | | | | | | Total #models = | 864 |

### A.7.2 How does the number of sub-models affect the ensemble?

Next we investigate how the number of sub-models change the performance of various deep ensemble models. Here, we choose DeepSVDD and VanillaAE as the baseline ensembles, for which results under 764 and 6,912 individual models are reported in Table 14, conducted on MNIST-4 dataset. We subsample the sub-models among all the 764 and 6, 912 models, with sizes equal to $[1, 5, 10, 15, 20, 30, 50, 100, 200, 500]$. We conduct each subsampling 100 times independently, and since we have 3 experimental runs reported in the previous sections of the paper, we provide the results regarding detection accuracy and variances among $3 \times 100$ experimental runs.

Fig. 5 shows the AUROC corresponding to different number of sub-models (for DeepSVDD we have both Polluted (left) and Clean (middle) settings, and for VanillaAE we have Polluted (right) setting only). Notice that when the number of sub-models is less than 20, the overall performance variance is relatively larger, where the ensemble performance is similar to an individual model prediction. AUROC quickly stabilizes and the variance shrinks as the number of sub-models becomes larger than 20 –especially for the DeepSVDD's Clean setting and VanillaAE's Polluted setting– with little difference beyond. These results suggest that ensembles are not sensitive to the number of sub-models beyond a certain size, where the larger the number, the more stable is the performance.

### A.7.3 HP Sensitivity Analysis: ROBOD

In this section, we perform the sensitivity analysis to both HP value ranges as well as the number of sub-models (since they are correlated) for our proposed ROBOD. We extend, shrink or shift the number of layers and number of BatchEnsemble models within the AE-S structure, which corresponds to changing the HP-ranges of NN widths and depths. We also recognize the additional training HPs such as train iterations, learning rate, dropout rate, weight decay, etc. Table 15 summarizes the HP-ranges we used to conduct our HP-sensitivity analysis based on Table 11. These additional experiments are on the Cardio dataset. For each set of HP configurations, we repeat 3 experimental runs and summarizes mean and variance of ROBOD's AUROC. The results among different hyper-ensembles (for both HP ranges and number of sub-models) are given in the following two tables, Table 16 (which alters the AE-S structure HPs) and Table 17 (which alters the other training HPs).

Our results on ROBOD match with the two observations in the previous subsections, in that ROBOD, the sped-up version of the i-ROBOD hyper-ensemble, provides stable results with the varying (1) number of sub-models, and (2) HP value ranges. Moreover, the mean performance and standard deviation of all ROBOD experiments listed in Tables 16 and 17 are considerably more competitive than all the other benchmarked models we studied, while ROBOD yields smaller variance to its own HP configurations than many banchmarked models (Table 7).

Our analyses provide practical insights on how to reduce an ensemble model's sensitivity to their HP settings. As long as time and resources permit, one should employ as many number of sub-models as possible, and expand the HP value ranges under a fine grid. In contrast, finding a single set of optimal/good HPs for an individual OD model is almost infeasible for the unsupervised setting.

Table 15: RobOD HP-ranges overview. Note that num_models corresponds to implicit ensemble over decay rate [1.5,1.75,2,2.25,2.5,2.75,3,3.25] (see Table 11), and num_layers corresponds to implicit ensemble over AE-i; e.g. num_layers=6 assembles over AE-2, AE-4, AE-6, AE-8, AE-10, and AE-12.

| List of HPs | Original Settings | Actions |
|---|---|---|
| BatchEnsemble num_models | 8 | Shrink:[4] Shrink:[5] Shrink:[6] Shrink:[7] |
| num_layers | 6 | Shrink:[4] Expand:[8] |
| Train iterations | [250,500] | Shift:[500,1000] Expand:[250,500,1000] Shrink: [250] |
| Train Learning Rate | [1e-3,1e-4] | Shift:[1e-1,1e-2] Expand: [1e-1,1e-2,1e-3,1e-4] |
| Dropout rate | [0.0, 0.2] | Shrink&Shift:[0.5] Expand: [0.0,0.2,0.5] |
| Weight decay | [0, 1e-5] | Expand: [1e-4,1e-5,0] Shrink: [0] |

Table 16: Results of RobOD over different AE-S structures. With different num_models and num_layers, various number of AE models are implicitly assembled. We report the mean AUROC and standard deviation across 3 runs, along with the number of (implicit) sub-models in parenthesis.

| **Mean&Std.** for different AE-S structures and the number of sub-models | | | | |
|---|---|---|---|---|
| | num_models:4 | num_models:5 | num_models:6 | num_models:7 |
| num_layers:4 | 92.4±0.6 (128) | 93.3±0.3 (160) | 93.1±0.4 (192) | 93.3±0.2 (224) |
| num_layers:6 | 93.6±0.4 (192) | 93.6±0.1 (240) | 93.5±0.1 (288) | 93.8±0.3 (336) |
| num_layers:8 | 93.8±0.1 (256) | 93.7±0.1 (320) | 93.5±0.1 (384) | 93.8±0.1 (448) |

Table 17: The altered HPs and results for RobOD. For example, "Train iterations: [250,500,1000]" means that the training iterations are expanded from the original experiment settings (see Table 15) to include the additional value [1000], while the AE-S HPs are the same as before (BatchEnsemble num_models equals to 8, num_layers equals to 6). Mean and Std. of AUROC over 3 runs are reported.

| Altered HP Ranges | **Mean&Std.** (RobOD) |
|---|---|
| No Changes (384 submodels) | 93.5±0.1 |
| Train iterations: [250,500,1000] (576 submodels) | 93.7±0.1 |
| Train iterations: [250] (191 submodels) | 94.0±0.2 |
| Train Learning Rate: [1e-1,1e-2] (384 submodels) | 93.7±0.2 |
| Train Learning Rate: [1e-1,1e-2,1e-3,1e-4] (576 submodels) | 93.8±0.0 |
| Dropout Rate: [0.5] (191 submodels) | 93.7±0.1 |
| Dropout Rate: [0.0,0.2,0.5] (576 submodels) | 93.6±0.2 |
| Weight Decay: [1e-4,1e-5,0.0] (576 submodels) | 94.0±0.0 |
| Weight Decay: [0.0] (191 submodels) | 93.7±0.3 |