# OpenReview forum: "Hyperparameter Sensitivity in Deep Outlier Detection: Analysis and a Scalable Hyper-Ensemble Solution"
_NeurIPS.cc/2022/Conference — NeurIPS 2022 Accept_

### Official Review · Reviewer_FAet · 2022-06-21

**Rating:** 6
**Confidence:** 5
**Soundness:** 3 good
**Presentation:** 4 excellent
**Contribution:** 3 good

**Summary:**

This paper studies the influence of hyperparameter for deep OD, as well as the strategy that alleviates the hyperparameter tuning issue during the training process. First, the authors soundly identifies the problem of hyperparameter selection in deep OD by extensive experiments with various AE-based deep OD methods and benchmarks. Then, with a general idea to avoid hyperparameter tuning via hyper-ensemble, the authors propose to leverage skip connection and batch ensemble technique to speed up the hyper-ensemble calculation. The results justify the effectiveness of the proposed strategy.

**Questions:**

1. The range of hyperparameters seems to be small in many experiments. Many hyperparameters only have two or three candidate values, which is not the case in practice. What will happen if the hyperparameter is selected from a larger range?

2. Is it possible to extend the strategy of ROBOD to other transformation based deep OD methods like [1]?

3. It will be better the conduct more extensive experiments on dataset like MNIST, not just the cases where 4/5/8 are used as inliers in Table 4. I am not sure why the authors pick those three cases to conduct the experiments. Can authors explain the reason?

Ref:

[1] Deep anomaly detection using geometric transformations. NeurIPS 2018.

**Limitations:**

Despite that this is a good work in general, authors should add more discussion on the limit of their work. No negative social impact.

**Strengths And Weaknesses:**

Strengths:

1. This is a well-motivated paper that tries to address a real problem that is often neglected. Hyperparameter selection is an important issue for almost all machine learning tasks, but it is especially challenging for unsupervised tasks like deep OD, as the outliers are often not accessible for validation. I appreciate the authors' effort to bring this topic up. Besides, the authors provides an extensive review on existing deep OD methods and a thorough analysis on them, which makes the discussion more convincing.

2. The presentation of the paper is good. I do enjoy reading this paper, as it follows a very clear pipeline: identifying a valuable problem-giving a possible solution-tailoring the solution for improvement, which is a good example that exhibits a complete process of research. Besides, all concepts are clarified and used consistently across the entire paper.

3. The proposed ROBOD seems to be simple and easy-to-implement, but enjoys both good effectiveness and efficiency against other AE-based baselines.

Weaknesses:

1.  The largest limit of this work is that the discussion seems to be limited to AE-based deep OD methods. I do understand that AE plays a center role in deep OD, but its performance is also shown by numerous works to be less satisfactory, especially on relatively complex datasets. By contrast, recent self-supervised deep OD methods (e.g. transformation based methods, CSI) achieve significantly better performance in many cases, and the authors did not discuss them as examples, which is a clear weakness to me.

2. The extendibility of the proposed ROBOD is a problem. For me, it is not straightforward to extend the skip connection based strategy to non-AE based deep OD methods that do not use a symmetric DNN architecture. In the meantime, ROBOD only considers the acceleration of two hyperparameters (though they are two important ones). For now, it is hard to call ROBOD a generally applicable framework for deep OD.

---

> ### Author Response · Authors · 2022-08-02
> **Rebuttal for Reviewer FAet**
>
> We would like to thank reviewer for their expert remarks on DeepOD and the insightful comments regarding our paper.
>
> **W1**. AEs are very popular, and have been heavily and successfully used for OD in the past [1,2]. Even the denoising autoencoders have been used for OD and can be seen as self-supervised deep OD, where generated noisy samples can be thought of as augmented data. Our sensitivity analysis is _not_ limited to AEs; in fact, we have picked one model from almost each family to analyze: AE, one-class, GAN, etc.
>
> We will be happy to add references to emerging self-supervised deep OD methods, like [3], to update our Related Work. On a separate note regarding self-supervised deep OD: We have ongoing work studying self-supervised OD; and in a gist, it turns out that the choice of the augmentation function/self-supervision heavily influences the results. In fact, and alarmingly, a poor choice makes the results worse than not doing any data augmentation. As such, type of self-supervision simply/merely remains to be a yet-another HP and is not an exception in that, carefully choosing it is important yet nontrivial. We are ready to publish those results soon, and would be happy to include a pointer here as soon as the draft is on Arxiv.
>
> **W2**. The speed up strategies are not all generally applicable, such as the skip connections being likely specific to AEs. However, please note that jointly-training multiple models under parameter-sharing and data sub-sampling are both general principles to speed up any ensemble model. The former has been used by BatchEnsembles (where the original model is not limited to AE architecture but general NN architectures with fully-connected layers, convolutional layers, etc.)  and the latter (sub-sampling) has been used by traditional (non-deep) OD models such as IForest. Our work points to a possible direction that hyper-ensemble can solve the HP sensitivity problem in deepOD, while ROBOD is the sped-up version for scalable training. In addition, our zero-masking is applicable to other deep OD architectures to ensemble various widths. As for different depths, techniques such as stochastic-depth neural network can be used instead of skip-connections. We will explore different architectures in our future work.
>
> **Q1.** What will happen if the hyperparameter is selected from a larger range? We conducted a HP-sensitivity analysis across different value ranges of HPs as well as different numbers of sub-models. Please refer to our updated Appendix Section 7. In a nutshell, we show that hyper-ensemble is robust to the number of sub-models beyond a certain size (~20-50), where the larger the number of sub-models, the more stable is the performance (Section A.7.2). In addition, Hyper-ensembles tend to achieve better stability to various value ranges of the HPs, while individual models are more sensitive and finding the optimal range is unreachable due to lack of validation (Section A.7.1). ROBOD, the sped-up version of the i-ROBOD ensemble, is also robust to its own HPs and exhibits considerably less variance than benchmarked algorithms (Section A.7.3). If one has no prior knowledge about the range of HPs, the best strategy would be including more HP ranges and training as many models as possible in the limited time frame.
>
> **Q2.** Is it possible to extend the strategy of ROBOD to other transformation based deep OD methods (like [3] below)?: Yes. As we discussed above: (1) jointly-training under parameter-sharing and (2) data sub-sampling are both general principles to speed up any ensemble model to increase its efficiency. For ensembling various widths, zero-masked layers with BatchEnsemble would also be broadly applicable.
>
> **Q3.** We agree that an even larger number of experiments (datasets/models/HPs/etc.) would strengthen our empirical conclusions, however owing to massive resource requirements, we could not provide all digits’ and image classes’ results due to numerous models to be trained with so many (hundreds of) different HPs on each new dataset. We would like to refer to [4], wherein Table 1 contains experimental results on MNIST digits and various CIFAR10 image recognition tasks. Digit “5”  and “8” are considered harder-to-train tasks, as some deep and non-deep classifiers provide poor results (i.e. generally lower AUROC). In comparison, for Digit “4” non-deep and deep classifiers provide similar results—which suggests that it can sometimes be challenging for deep OD models to achieve better performance than non-deep models. In general, our experiments show that deep OD models can perform worse than non-deep models due to HP settings, regardless of which dataset we choose. Rest assured, neither our datasets nor the investigated models are “cherry picked” and we will be ready to do any specific additional experiments requested.

---

> > ### Author Response · Authors · 2022-08-02
> > **Rebuttal for Reviewer FAet - PART2**
> >
> > In conclusion, we proposed a parameter-sharing AE ensemble toward creating a deep OD model that is more robust to HPs.  Ensembles are costlier to train. Hence, there exists a robustness (and associated expected accuracy) versus time tradeoff. We have proposed as new contributions three (3) different ways to speed up the ensemble training to reduce the computation/memory overhead, namely: 1) skip-connections for depth ensembling; 2) batch ensemble and zero-masking for width ensembling; and 3) sub-sampling.
> >
> > We think the main limitation is that the time and resource requested for the ROBOD is larger due to its ensemble nature. In sacrifice of training time, we reduce the model’s dependency on the hyper-parameters and successfully reduce the search effort. Regardless of ensembling or individual models, they all need a range of HPs to start with. In order to find a reasonable range of HPs, one possible solution is to utilize the auto-ML techniques discussed in [5], which can further reduce the search effort.
> >
> >
> > *[1] Charu C. Aggarwal and Saket Sathe. 2017. Outlier Ensembles: An Introduction (1st. ed.). Springer Publishing Company, Incorporated.*
> >
> > *[2] Chalapathy, Raghavendra & Chawla, Sanjay. 2019. Deep Learning for Anomaly Detection: A Survey.*
> >
> > *[3] Deep anomaly detection using geometric transformations. NeurIPS 2018.*
> >
> > *[4] Ruff, L., Vandermeulen, R., Goernitz, N., Deecke, L., Siddiqui, S.A., Binder, A., Müller, E. Kloft, M. 2018. Deep One-Class Classification.*
> >
> > *[5] Zhao, Yue, Rossi, Ryan and Akoglu, Leman. 2021. Automatic Unsupervised Outlier Model Selection.*

---

> > > ### Author Response · Authors · 2022-08-08
> > > **Additional Notes**
> > >
> > > Dear Reviewer FAet,
> > >
> > > We would really appreciate if you could have provide some responses to our comments, since there is only one day left from the discussion period. We would love to address any further questions and concerns you have. Please let us know if our clarifications are useful in any way.
> > >
> > > Thank you very much for your time!

---

### Official Review · Reviewer_9hEy · 2022-07-13

**Rating:** 7
**Confidence:** 3
**Soundness:** 3 good
**Presentation:** 3 good
**Contribution:** 3 good

**Summary:**

The papers provide 2 main contributions:

1. The paper performs a large-scale study of how different model hyperparameters (e.g., number of layers, number of training iterations, learning rate, etc.) affect the performance of several existing deep learning-based outlier detection models (vanilla autoencoder, RDA, DeepSVDD, GANomaly, and RandNet). The results show that these models are quite sensitive to the hyperparameter choices, which are often not explored in the original papers. Furthermore, ensembling the outputs of these models (across different hyperparameter runs) generally yields higher outlier-detection performance than methods for choosing a single hyperparameter.

2. The paper proposes an "ensemble" of autoencoders of varying depths and widths called RobOD, which is used for outlier detection. I put "ensemble" in quotes, because RobOD is not an ensemble of independently trained models, but rather an "ensemble" in a similar sense to using Monte-Carlo dropout to generate a large set of models from a single trained model. Empirical results demonstrate how RobOD is competitive at outlier detection against other models, while being significantly more robust to the hyperparameter choices. (An alternative interpretation is that RobOD eliminates having to choose between depth and width hyperparameters, so there are simply fewer hyperparameters to deal with.)

**Questions:**

**Q1: What exactly is the "polluted" setting?** \
My understanding is that in the "polluted" setting, you have a training set that includes both inliers and outliers, but you don't actually have the labels that separate the inliers from the outliers. Is my understanding correct?

**Q2: How well does the RobOD style of ensembling across different model depths and widths apply to autoencoder tasks in general, beyond outlier detection?** \
Essentially, I'm struggling a bit to understand whether the authors think that managing hyperparameters is unique to (or particularly difficult with) outlier-detection, which I don't think is the case. I think the authors could have a much stronger paper if they demonstrate how their multi-depth / multi-width autoencoder setup improves the performance of autoencoders in general. For example, did this ensembling reduce reconstruction loss?

**Limitations:**

The authors briefly discuss that their method is costlier than training a single model, and they do not list any negative societal impacts.

**Strengths And Weaknesses:**

## Strengths

**S1. Large-scale experiments**\
Through a large set of experiments, the authors conclusively demonstrate the importance and influence of hyperparameters on current deep learning-based outlier detection models.

**S2: RobOD demonstrates a method for creating an "ensemble"-like autoencoder model"** \
By enforcing weight-sharing and adding reconstruction losses at varying depths, RobOD only requires a single autoencoder model that can "simulate" multiple autoencoders of varying depths and widths.

## Weaknesses

**W1. Missing details on how autoencoder is used for OD**\
Perhaps this comes from my ignorance about OD models in general, but in the paper I didn't see any mention of how an autoencoder model is actually used for outlier detection. Suppose I have trained an autoencoder in an unsupervised manner by minimizing reconstruction loss. Then what? How is it used for outlier detection? For example, I can envision several ways of using a trained autoencoder for OD. If the autoencoder is a variational autoencoder (VAE), then I can calculate an estimated density of a given example under the training data distribution. If the autoencoder is deterministic, then I could threshold the reconstruction loss: if an example has high reconstruction loss, then I deem it an outlier. Are you using either of these methods? Or something totally different?

**W2. Missing analysis of RobOD ensembling procedure**\
One thing I'd like to see is whether the multi-depth / multi-width autoencoder setup improves the performance of autoencoders in general. For example, did this ensembling reduce the autoencoder's reconstruction loss?

**W3. Does not separate variation in hyperparameters from variation in initialization**\
Maybe I missed this point somewhere in the text, but I struggled to separate the two main sources of variation in model output. First, there are many hyperparameter choices. Second, there are many random initializations. How are the authors certain that the benefit from "ensembling" was due to ensembling across hyperparameters, instead of ensembling across many random initializations? If you took a vanilla autoencoder with the same width and depth as the widest/deepest RobOD constituent, then used many random initializations, how does that compare against RobOD?

**W4. Missing contextualization with key related works**\
To me, the multi-width ensembling adopted in RobOD is quite similar to Monte-Carlo dropout, except that the dropping-out is not random but rather follows a particular deterministic pattern. If my understanding of RobOD is incorrect, please correct me! However, if my understanding is correct, then I think it's worth addressing this similarity. Likewise, the multi-depth ensembling in RobOD is quite similar to stochastic-depth models, except again the dropping-out of layers is not random but follows a particular deterministic pattern. Furthermore, the RobOD autoencoder also somewhat resembles a "U-net," except that the loss is reconstruction instead of segmentation. Making these connections to existing literature would be helpful to understanding RobOD.

---

> ### Author Response · Authors · 2022-08-02
> **Rebuttal for Reviewer 9hEy**
>
> We want to thank the reviewer for their valuable questions and insightful observations in our paper.
>
> First, we kindly correct and clarify that ROBOD is a **typical** ensemble model of parameter-sharing AEs. In particular, ROBOD assembles multiple (what we call) AE-S models (where S refers to Skip-connection). An AE-S is a single model that we designed to assemble two HPs under-the-hood/implicitly: 1. depth and 2. width (via parameter-sharing). For the other base model HPs (such as learning rate, weight decay, etc.), we train *multiple* AE-S models with various configurations independently. Thus, ROBOD remains to be a typical ensemble model of multiple AE-S models, each of which yields multiple parameter-sharing AEs (again, each single AE-S in fact being multiple AEs of varying width and depth).
>
> i-ROBOD provides an explicit dropping of the neural network (NN) depth and width, while MC-dropout attempts to randomly dropout the nodes in each layer. The NN with MC-dropout _cannot_ be considered as shrinking the widths and depths in the NN. An ensemble NN that is similar to MC-dropout, as we mentioned in the paper, is RandNet (citation 11 in the paper). However, our experiments have shown that even RandNet cannot achieve results that are in-sensitive to different choices of HPs.
>
> **W1.** To answer how the autoencoder is used for OD: Yes, we are using the reconstruction loss averaged from all based models in the ROBOD ensemble as the outlier score. Utilizing the reconstruction loss of the autoencoder (low: normal, high: outlier) has been a widely-used approach in the DeepOD community. In the static data setting, samples can be ranked from most to least anomalous by this score. In the dynamic setting, one may use a threshold to filter outliers, where techniques exist in the literature for dynamically identifying a good threshold. However, one can foresee the difficulty to train such autoencoders, especially in the unsupervised setting, where autoencoders with sufficient capacity/complexity may fit well (overfit) with low reconstruction even to outliers.
>
> **W2.** We will provide an update on our experimental results with reconstruction loss. Our short answer is that reconstruction loss is not necessarily lower for ROBOD. Due to parameter-sharing (a constraint), in some cases each model may become sub-par in terms of fitting to training data (i.e. higher training/reconstruction loss). On the other hand, in other cases, such constraints may serve as implicit regularization, which may also prevent an extremely high train/reconstruction loss due to bad initialization. In fact, one may argue that such regularization helps ROBOD outperform i-ROBOD  (the non-sharing/independent base model training version) in OD performance. We are interested in the theoretical analysis of this phenomenon, but it is out of the scope of our paper.
>
> We also argue that the absolute numbers of reconstruction loss cannot be solely used to evaluate whether ROBOD (or any AE model really) performs well for OD tasks. Due to the nature of unsupervised outlier detection settings, the data can be polluted (i.e. outliers are mixed into the inlier data), and therefore, AE is prone to overfit the data (including outliers (!)) in general. Low reconstruction loss does not guarantee a good model fit, when outliers also have low reconstruction errors.
>
> **W3**. How are the authors certain that the benefit from "ensembling" was due to ensembling across hyperparameters, instead of ensembling across many random initializations:  We think it is an extremely good and important question. In fact, one of the baseline models we are comparing to, RandNet [1] is ensembling several VanillaAEs (with the same/fixed HP) over random initializations (i.e. it is _not_ a _hyper_ ensemble). Our preliminary results in Figure 9 and Table 4 have shown that even if one assembles #10-100 sub-models with the same HP setting, across different random initializations, one still gets a higher variance than the hyper-ensembles.
> It is well known in the ensemble learning literature that diverse models that make uncorrelated errors are what makes an ensemble superior to its individual constituents/parts. Here, one can think of using different HPs as a way of introducing diversity to our (OD) ensemble. Similarly, random initializations also provide diversity and as such they also likely provide gains in performance. However, recall that our ultimate goal is not (necessarily) high performance – but rather, a HP-robust OD model. As such, using various different HPs is essential/a must-do for our design but we mainly do so to reduce the sensitivity to HPs, although it does also come with added diversity and subsequently often improved performance (as a byproduct of our design).

---

> > ### Author Response · Authors · 2022-08-02
> > **Rebuttal for Reviewer 9hEy - PART 2**
> >
> > **W4.**  Perhaps at a fundamental level, model ensembling, MC dropout, stochastic-depth models are ideas around _diversifying_ ensemble models. In ensemble learning, diversity is to improve performance, owing to uncorrelated errors being canceled out. In MC dropout, the diversity prevents overfitting to training data and improves generalization performance. If the reviewer agrees that perhaps the common thread is diversity (randomized or deterministic), we will be happy to update our Introduction and Related Work sections to include these connections. One major difference of our work from MC dropout, stochastic-depth, and model ensembling in general is the purpose of “diversifying”: we reiterate that while high performance is important, it is secondary to us. The primary reason we design a _hyper_ ensemble is because we want to resolve/circumvent the bottleneck of selecting the “best” hyper-parameters and reduce the detection sensitivity (i.e. performance variation) due to hyper-parameters. Separately, we remark that MC dropout cannot be regarded as an ensemble of deterministic different width neural networks. For BatchEnsemble neural networks, which is our main idea for speeding up the ensemble training, MC dropout or stochastic-depth ideas are more difficult (nontrivial) to apply for hyper-ensembling width and depth, due to their random nature.
> >
> >
> > **Q1.** On what exactly the "polluted" setting is: The reviewer’s understanding is correct. In the polluted setting, training and test data are the same and labels are not available (that is why it is also called fully unsupervised). We train on the inliers and outliers together (without knowing which point is which) and then test the same set of points against the trained model. Compare this to the non-polluted setting, named Clean (or Disjoint setting): the training and test data are disjoint, where training data is inliers-only (hence known labels), and test data consists of (unlabeled) set of outliers and another set of inliers (different and disjoint from those in the training data).
> >
> > **Q2.** On how ensembling across different model depths and widths apply to autoencoder tasks in general, beyond outlier detection and whether we think that managing hyperparameters is unique to (or particularly difficult with) OD:  Yes, we do believe managing HPs is _especially difficult for unsupervised ML tasks_ as there is no guidance in how to choose them. This is unlike using something like validation/hold-out data (with labels) / cross-validation approach for supervised problems that guide toward the generalization performance of a model (and its HPs).  We believe that ROBOD ensembling can be applied to various unsupervised tasks, which we will study in the following works. Outlier detection is one important domain of unsupervised learning tasks. We agree that the ensemble design we have proposed can readily be used for supervised problems as well, where one could study the effect of parameter-sharing on training performance (e.g. reconstruction loss) as well as its regularization effect on generalization, which however are not in the scope/purpose of our paper.  Our primary goal in parameter-sharing in our designed ensemble is _speed up/efficiency_, i.e. to reduce the computational and memory overhead of training independent base models in the ensemble while achieving higher robustness to HPs.
> >
> > In conclusion, we proposed a parameter-sharing AE ensemble toward creating a deep OD model that is more robust to HPs.  Ensembles are costlier to train. Hence, there exists a robustness (and associated expected accuracy) versus time tradeoff. We have proposed as new contributions three (3) different ways to speed up the ensemble training to reduce the computation/memory overhead, as reviewer 1 above outlined, they are: 1) skip-connections for depth ensembling; 2) batch ensemble and zero-masking for width ensembling; and 3) sub-sampling.
> >
> > We do not foresee any negative societal impacts of our model. Ours is an empirical evaluation and methodological innovation paper. Outlier detection is widely used for many real world applications such as fraud, malice, fault, defect, etc. detection in finance, computer and network security, manufacturing, etc. We have studied HP-sensitivity of a family of deep OD models within the field and proposed a new ensemble model that adds to this literature.
> >
> >
> > *[1] Chen, Jinghui & Sathe, Saket & Aggarwal, Charu & Turaga, Deepak. (2017). Outlier Detection with Autoencoder Ensembles. 10.1137/1.9781611974973.11.*

---

> > > ### Author Response · Authors · 2022-08-08
> > > **Updates for W2- experimental results**
> > >
> > > We are thankful for the reviewer's valuable suggestions, and we would like to provide some updated results we have for the ensembling effects, regarding the reconstruction loss (answering **W2**).
> > >
> > > Here we fix the hyper-parameter ranges for the ROBOD and AE models, with the same configurations as our paper's Experiment section (refer to Table 9 and Table 10). We calculate the average reconstruction loss of ROBOD and AE models,  *per each individual sample*, regardless of whether it is an outlier or not. We show the average reconstruction loss across several datasets, and the results are shown as below:
> > >
> > > |       | MNIST-4 | MNIST-5 | MNIST-8 | CIFAR10-0 | CIFAR10-1 | Thyroid | Cardio | Lympho |
> > > |-------|---------|---------|---------|---------|---------|---------|--------|--------|
> > > | ROBOD | 4.33    | 4.89    | 4.67    | 2.09    | 5.91    | 0.08    | 1.22   | 2.24   |
> > > | AE    | 3.45    | 3.07    | 3.91    | 7.05    | 2.35    | 0.15    | 0.63   | 2.69   |
> > >
> > > Out observation is that ROBOD has a larger average reconstruction loss for MNIST data, CIFAR10-1 and Cardio. However, ROBOD's reconstruction loss is smaller for CIFAR10-0, Thyroid and Lympho data. This experiment results are consistent with our previous rebuttal for  **W2**:   the reconstruction loss is not necessarily lower for ROBOD. In some cases model can become sub-par due to parameter-sharing. While on the other hand, the parameter-sharing plays as implicit regularization. Therefore, the parameter-sharing becomes both a constraint to prevent overfitting and extremely high loss from bad initialization.
> > >
> > > This experiment also illustrates that reconstruction loss is not directly related to the performances of deep OD models under unsupervised setting. Some deep OD models may yield low reconstruction loss due to overfitting, and thus lack the ability to detect outliers.

---

> > > > ### Author Response · Authors · 2022-08-08
> > > > **Additional Notes**
> > > >
> > > > Dear Reviewer 9hEy,
> > > >
> > > > We would really appreciate if you could have provide some responses to our comments, since there is only one day left from the discussion period. We would love to address any further questions and concerns you have. Please let us know if our clarifications are useful in any way.
> > > >
> > > > Thank you very much for your time!

---

> ### Comment · Reviewer_9hEy · 2022-08-10
> **Feedback to Author Response**
>
> Thank you to the authors for answering my questions. I particularly appreciate the new experiments on the muti-width / multi-depth autoencoder, which show that reconstruction loss is not always improved with the multi-width / multi-depth design. I also appreciated the clarification about why sensitivity to hyperparameters in the OD setting is possibly more challenging than in supervised learning. The new experimental results added to the Appendix also convincingly demonstrate that RobOD is reasonably robust to its own choices of hyperparameters.
>
> As my questions have been adequately addressed, I am happy to raise my score from 5 (Borderline accept) to 7 (Accept).

---

### Official Review · Reviewer_9Tjo · 2022-07-13

**Rating:** 5
**Confidence:** 3
**Soundness:** 2 fair
**Presentation:** 2 fair
**Contribution:** 2 fair

**Summary:**

This paper studies hyperparameter (HP) sensitivity for the outlier detection (OD) task, which is a fundamental challenge in the unsupervised setting. The authors conducted an extensive empirical study of several deep OD models' performance with various HP choices and demonstrated these SOTA OD models are very sensitive to HP selection. A deep auto-encoder ensemble model is proposed to reduce the dependency on HP and achieve better OD performance with the help of the ensemble. In order to prevent long running time for training each individual model in the ensemble, model parameters are shared among multiple models through three techniques: 1) skip link for depth ensembling; 2) batch ensemble for width ensembling; and 3) sub-sampling. Several experiments have been performed to demonstrate the effectiveness of the proposed ensemble method and the efficiency of the model training.

**Questions:**

1. The reviewer wonders if the authors can provide the analysis of the HP robustness of the ensemble methods. For example, try multiple different settings in Table 9, resulting in various numbers of sub-models, to demonstrate the HP robustness of the ensemble methods.
1. Table 2 gives the **Hyper-ens. Mean** and **Mean**. Although the performance seems improved here, the stddev for the **Hyper-ens. Mean** is generally higher than its counterpart. Does this mean that the ensemble model here is also sensitive to HP?
1. Although the parameter sharing over multiple deep AEs helps with the training speed, the reviewer guesses that such a strategy will degrade the outlier performance for each sub-model (the losses for sub-models are competing with each other) resulting in overall lower performance for the entire ensemble, compared to training each model individually. However, in Table 4, it seems ROBOD outperforms i-ROBOD in multiple tasks. Any reasoning would help the reviewer understand how parameter sharing helps with OD accuracy performance.
1. It seems IF presents good results over vector-based data, and even for CIFAR10-auto. Any comparison/analysis between deep models vs. traditional OD methods would help readers understand the difference.
1. For image task, does ROBOD use CNN architecture, or FC?

**Limitations:**

The main limitation of this paper is mentioned as the second weakness previously. If the major issue (HP robustness of the ensemble ) is not answered, the use of the ensemble method cannot be justified, which also weakens the training speed-up claim for the ensemble method.

**Strengths And Weaknesses:**

* Strengths
  1. This paper raises an essential question for the outlier detection task. Under an unsupervised setting, the model performance is agnostic before actual prediction/testing due to the lack of ground-truth label information, resulting in no clear HP selection strategy, compared to the supervised counterparts (validation).
  1. The reviewer appreciates the time and efforts put into the experimental study of the existing OD models in Section 3. This study clearly points out that most deep OD models are sensitive to HP selection, and the final performance of models may suffer from potentially-large variation.
  1. The parameter-sharing AE model and the three related ensemble techniques proposed in this paper may help reduce the training time for large deep ensembles.
* Weakness
  1. Although rarely used in deep OD models (majorly due to long training time), the ensemble ideas are widely used in OD tasks. For example, the Isolation Forest (IF) is an ensemble tree method. The ensemble ideas inside this paper simply aggregate the outlier scores from models inside the ensemble, which is the common practice.
  1. The paper doesn't actually answer **RQ1) how to design an unsupervised deep OD model that is robust to its HPs**. Note that **RQ1** is different from **Q1** and **Q2** given in Section 5. Although the ensemble method reduces the HP dependency for each individual sub-model inside the ensemble, the HP for the entire ensemble is not studied. For each sub-model, the HP may be model depth, layer width, learning rate, etc. The HP for the entire ensemble may be the number of sub-models, sub-model HP selection range, etc. The first paragraph in Section 4.2.3 gives an example of the sub-model HP selection range. The current ensemble presented in the paper (Table 9 in the appendix) can only be considered as a single HP setting for the entire ensemble. It is unclear how the method performs if different values are given for Table 9. Therefore, we cannot tell if the proposed ensemble is robust to its HPs or not (RQ1).

---

> ### Author Response · Authors · 2022-08-02
> **Rebuttal for Reviewer 9Tjo**
>
> First, we thank the reviewer for their suggestions and valuable questions for us to make improvements on our paper.
>
> **W1**: Regarding that ensembles are widely used and simply aggregate the outlier scores from models inside the ensemble, we recognize that model ensembling is a common practice. However, the purpose of our ensemble is novel; we try to reduce the sensitivity of deep models to their HPs (the first one to our knowledge). Put differently, not all ensembles are hyper-ensembles. Of traditional ensemble OD models, including IsolationForest, the ensemble is not necessarily to improve the robustness of the model to HPs. Moreover, within the ensemble framework, our main contribution is the speed up/efficiency strategies, which is particularly challenging for the deep models that are cumbersome to train [1] as recognized by the reviewer. We also see the potential to extend the speed-up/efficiency ideas we have proposed to other ensemble models (i.e. beyond AEs).
>
> **W2** and **Q1**. To answer **RQ1) how to design an unsupervised deep OD model that is robust to its HPs.** We provide additional experimental results in Appendix A.7 of our updated paper. For verifying the robustness of HP-ensembles with respect to its HPs, we follow the reviewer’s advice and conduct experiments with ROBOD, i-ROBOD, DeepSVDD ensembles, with different number of sub-models and ranges of HP selections. In short, our observations are:
> * Ensembles are not sensitive to the number of sub-models beyond a certain size (~20-50), where the larger the number of  sub-models, the more stable is the performance (Section A.7.2).
> * Hyper-ensembles tend to achieve better stability to various value ranges of the HPs, while individual models are more sensitive and finding the optimal range is unreachable due to lack of validation (Section A.7.1).
> * ROBOD, the sped-up version of the i-ROBOD ensemble, is also robust to its own HPs and exhibits considerably less variance than benchmarked algorithms (Section A.7.3).
>
> By the above additional experiments, we have also answered the reviewer’s **Q1** about whether the ensemble ROBOD is robust to multiple different settings in Table 9 and different number of sub-models. The results conclusively show that ensemble models achieve lower variation in performance. We thank the reviewer again for creating the opportunity for us to strengthen our paper w.r.t. HP-robustness.
>
> **Q2**. The stddev for the Hyper-ens. Mean is generally higher than its counterpart. Does this mean that the ensemble model here is also sensitive to HP? The short answer is that these two std.dev. ’s are not directly comparable. Let us first clarify the definitions and then elaborate.
>
> In Table 2, **Mean (&Std)**  depicts the *average of AUCs* (& their stddev) across models with different HP settings, and  **Mean (avg. 3 runs)**  shows the average of the said Mean across 3 separate runs (i.e. initializations). As such, **Mean (avg. 3 runs)** is the empirical expected value of the Mean. On the other hand, **Hyper-ens. Mean (avg. 3 runs)** reports the average performance of the Hyper-ens. across 3 separate runs, where the performance is the AUC of the scores averaged across models with different HP settings. (Note the difference: Here we are averaging scores across models and reporting a single AUC of this average score, rather than averaging the AUCs of the individual models as with the Mean above.)
>
> As such, the two std.dev.’s are not directly comparable: the former is the std. associated with the Mean of Mean AUCs, whereas the latter is the std. of the Hyper-ens. performance. What is comparable is the *Std. associated with the Mean* versus the *Std. associated with the Hyper-ens*. *Mean. Std associated with the Mean* depicts the variation in performance by using one HP setting (i.e. a single model), whereas the *Std. associated with the Hyper-ens. Mean* shows the variation in performance of the Hyper-ensemble, which is almost ***two orders of magnitude smaller***, clearly showcasing the robustness of ensembling.
>
> **Q3**: Our initial conjecture was also that individual models would be relatively sub-par due to the “implicit constraints” on parameters due to sharing. However, it turns out that parameter-shared models assembled together remain to perform well or even better than non-shared ensemble i-ROBOD (i.e. the reviewer’s observation is accurate).  An explanation of this outcome may be the regularization effect of parameter-sharing. Parameter-sharing has been known to provide a good regularization effect to initializations in CNN training [2]. Implicit constraints on the parameters likely prevent them from overfitting (to noisy inliers and also to existing outliers in training data under the Polluted setting). More detailed analysis should be done in theories and experiments, which is outside the scope of this paper. We refer the reviewer to Appendix Section A.4, which describes the effect of parameter-sharing within the AEs.

---

> > ### Author Response · Authors · 2022-08-02
> > **Rebuttal for Reviewer 9Tjo - PART 2**
> >
> > **Q4**. Regarding the good results of IF over vector-based data, and comparison between deep versus traditional OD methods: IF does well fairly often, and has been shown to outperform numerous other popular (non-deep) OD models in large benchmark studies [4, 5(see Table 1), 6(see Table IX)] on vector-datasets. Being an ensemble, it is robust to HPs; whereas, as we argue, for (individual/single) deep models to “catch up” they should be HP-tunable. Under “good” HPs, deep models outperform IF significantly. This is true particularly for image datasets, where representation learning matters to a much larger extent as compared to vector data where the features may not be as correlated/interdependent spatially. However, under “bad” HPs and even on average, deep models can perform poorly and be undesirable against IF.
> >
> > This is exactly the point we make in our paper: The glory of deep models is subject to a good HP choice (which is nontrivial), and that traditional models may be more preferable on average—until the pressing HP-tuning problem is systematically addressed for deep OD models. Our paper is the first “wake up call” about the HP-sensitivity of the deep models; shining the spotlight on the important unsupervised model selection (UMS) problem.
> >
> > Beyond this “wake up call” to the ML community to work on the important UMS problem (for OD, or more broadly unsupervised representation learning, or unsupervised clustering tasks), we proposed one possible ensemble solution. Unlike traditional, relatively fast ensemble models like IF which are efficient to train, HP-ensembling deep models is cumbersome—to which end, we also proposed various speed-up strategies. We believe the two-pronged contributions of our work (1. a large-scale sensitivity analysis that objectively demonstrated the pressing issue, and 2. a new, sped up HP-ensemble design) are timely and important contributions to the unsupervised learning literature at large, and outlier mining community specifically.
> >
> > **Q5**. For image task, does ROBOD use CNN architecture, or FC? ROBOD currently only uses FC layers since we have not found a speeding-up strategy that can employ the CNN architecture. However, we believe it would be a challenging and valuable future direction as CNNs may further help improve detection performance for image data.
> >
> > To conclude, we would like to remind the reviewer that ROBOD (proposed ensemble model) constitutes only (the second) half of our paper, and that our large-scale HP-sensitivity analysis of deep OD models itself is the first such analysis for deep models, that comprises the other half of our work. Many deep OD papers have been written that report superior performance to traditional models. However, almost none of them brings up the HP-tuning challenge, nor provides sufficient details on how HPs were tuned exactly for their reported numbers. We believe that our paper 1) reveals/documents their HP-sensitivity objectively, and 2) “rings the alarm bells” regarding the issue–that the community should be aware and transparent of this issue. We bring to the spotlight that unsupervised model selection (UMS) needs to be tackled systematically and prioritized for new research. Put differently, developing yet-another deep OD model cannot be the top priority before UMS is tackled, as deep models cannot reach their often-exaggerated “glorious” performance without proper HP-tuning.
> >
> > *[1] Charu C. Aggarwal and Saket Sathe. 2017. Outlier Ensembles: An Introduction (1st. ed.). Springer Publishing Company, Incorporated.*
> >
> > *[2] Ian Goodfellow, Yoshua Bengio, and Aaron Courville. 2016. Chapter 7.Deep Learning. The MIT Press.*
> >
> > *[3] Steinbuss, Georg and Klemens Böhm. 2021.Benchmarking Unsupervised Outlier Detection with Realistic Synthetic Data.ACM Transactions on Knowledge Discovery from Data (TKDD) 15 (2021): 1 - 20.*
> >
> > *[4] Emmott, A. F., Das, S., Dietterich, T., Fern, A., & Wong, W. K. (2013, August). Systematic construction of anomaly detection benchmarks from real data. In Proceedings of the ACM SIGKDD workshop on outlier detection and description (pp. 16-21).*
> >
> > *[5] Emmott, Andrew, et al. "A meta-analysis of the anomaly detection problem." arXiv preprint arXiv:1503.01158 (2015).*

---

> > > ### Author Response · Authors · 2022-08-08
> > > **Additional Notes**
> > >
> > > Dear Reviewer 9Tjo,
> > >
> > > We would really appreciate if you could have provide some responses to our comments, since there is only one day left from the discussion period. We would love to address any further questions and concerns you have. Please let us know if our clarifications are useful in any way.
> > >
> > >
> > > Thank you very much for your time!

---

> > ### Comment · Reviewer_9Tjo · 2022-08-09
> > **Feedback to Authors' Rebuttal**
> >
> > Thanks a lot for clarifying these questions, which helps the reviewer understand the paper better, especially for **Q2** and **Q3**. The reviewer understands that this paper is a "wake-up call" for the unsupervised model selection problem, and proposed approaches to improve the computation efficiency of ensemble AEs training. The reviewer really appreciates the authors' efforts in conducting extra experiments to demonstrate hyper-ensembles' robustness against HPs in Appendix A.7. Based on these, the reviewer would like to rescore my overall rating from 4 (Borderline reject) to 5 (Borderline accept). Here is the major concern on why the reviewer doesn't give a higher score than the current rating.
> > * Although additional experimental studies were conducted to address **Q1**, the reviewer is still concerning about whether **RQ1** is actually answered here.
> >   * From a theoretical perspective, there exist two questions to be answered for ROBOD (or ensemble methods in general): 1) Why does the ensemble help with the robustness of its sub-model against its sub-HP? 2) Why is the ensemble robust against its own HP (e.g. sub-HP range)? Some of the experiments in the original paper and rebuttal are conducted to answer these questions; however, none of these questions are fully reasoned or theoretically analyzed in the paper. Only using empirical studies to demonstrate the effectiveness of the proposed methods makes it less sound.
> >   * From an experimental perspective, the current experiments are not enough to fully demonstrate if ROBOD (or i-ROBOD) is robustness against its HP. The empirical study in Table 12 gives the performance of i-ROBOD under different sub-HP ranges. For example, under the rows of *Number of encoder layer:[6,7,8,9]* and *Weight decay: [1e-3,1e-4]*, the performance significantly drops compared to other i-ROBOD experiments. Since the performance of the model is agnostic before the actual testing for unsupervised outlier detection, there still exists chance that the hyper-ensemble will perform badly due to bad HP choice. Probably the chance of selecting bad HP is lower than the individual model (w.o. ensembles), but it is hard to conclude that ROBOD is robust to its HP in general. Moreover, the current experiments (in Appendix A.7) are conducted for two small datasets (MNIST and Cardio), the reviewer wonder if similar observation can be given over larger deep models for other (large) datasets.

---

> > > ### Author Response · Authors · 2022-08-09
> > > **RE:  Feedback to Authors' Rebuttal**
> > >
> > > We thank the reviewer again for the valuable questions. The reviewer's suggestions are very helpful as we choose the topics of our future work.
> > >
> > > To answer, *1) Why does the ensemble help with the robustness of its sub-model against its sub-HP?*  We are proposing several future directions to study this phenomenon:  (a) If each sub-model is drawing a part of the actual decision boundary, a good ensemble may be capturing a more complete image of the boundary than each sub-model. (b) The improvement of ensemble may relate to average of outlier scoring. Outliers can stand out with better chances, if detected by some member models. As none of the literatures in Deep Outlier Detection have explained such phenomenon, we hope to relate our ideas to ensembles in out-of-distribution learning and other areas, for example, Kumar *et. al.* ’s calibrated ensemble work [1].
> > >
> > > For *2) Why is the ensemble robust against its own HP (e.g. sub-HP range)?* Due to time constraint, we are not able to showcase the ROBOD performances for all datasets, but we would like to add additional experiments in the future. In our current experiments, we have shown that Hyper-ensembles tend to achieve better stability to various value ranges of the HPs, and it provides significant improvement with stability than other single model. We agree with the author that ROBOD's performance can drop if a bad range of HPs are selected. That is why we provide a guideline for ROBOD training: with limited resources one may want to train with as many configurations as one can, such that the probability of including a reasonable range of HPs is high. Additionally, one could refer to existing works [2] for the HP range selection problem.
> > >
> > >
> > > *[1] Kumar, Ananya, et al. "Calibrated ensembles can mitigate accuracy tradeoffs under distribution shift." The 38th Conference on Uncertainty in Artificial Intelligence. 2022.*
> > >
> > > *[2] Zhao, Yue et al. “Automatic Unsupervised Outlier Model Selection.” 2021.*

---

### Author Response · Authors · 2022-08-08
**Updated Results and Feedback Welcomed**

Hello Reviewers and Area Chair,

We have updated our paper’s appendix (Section A.7), in order to address Reviewer 9Tjo  and Reviewer FAet’s questions about ROBOD’s robustness for its own hyper-parameters. Specifically, we study the two hyper-parameters that 9Tjo requested: (1) the number of sub-models and (2) the selection of hyper-parameter ranges. In addition, we provide results on the general hyper-ensemble’s robustness, while we argue that the hyper-ensemble method can be extended to various types of DeepOD models. We also address Reviewer 9hEy’s concerns about the ensembling performances, deep outlier detection objectives, etc.

We want to thank the reviewers for all the suggestions, and would like to hear back if you have more questions regarding our paper. If our responses have addressed your questions, we would kindly ask for your reconsideration of the scores.

---

### Meta-Review · Area_Chair_aAMB · 2022-08-28

**Recommendation:** Accept
**Confidence:** Certain

**Metareview:**

This paper empirically demonstrates the sensitivity of unsupervised OD methods to hyperparameters and proposes ensembles of models with differing hyperparameters along with training techniques based on weight sharing to do so efficiently.

The authors provided additional experiments to answer some reviewer's major concerns regarding the (meta-)HP robustness of the ensemble methods. While there are natural fluctuations with respect to the (meta) hyperparameters, a larger number of hyperparameters included usually resulted in a close to optimal AUROC.

Another concern for two reviewers was the extendability of the efficient ensemble training techniques to other models such as GANs etc. As the authors replied, even though skip connections might not be adaptable to other techniques, e.g. for ensembling various widths, zero-masked layers with BatchEnsemble would also be broadly applicable to ensemble-learn other unsupervised representation learning models. Perhaps in the final version, the authors further discuss with a short experiment how the AE-specific scaling techniques add to the performance.

A further concern was the lack of theoretical underpinning about the (meta)HP-robustness. Given that the reviewers agree that this paper provides ample empirical evidence and praise the experimental value, we think theoretical work (providing HP sensitivity results is highly nontrivial in general, let alone for neural networks) can be part of future work but the lack of it in the current manuscript should not prevent the publication of this work.

**Award:**

No

---

### Decision · Program_Chairs · 2022-09-14

Accept